# What climate signal is contained in decadal to centennial scale isotope variations from Antarctic ice cores?

Thomas Münch[1,2] and Thomas Laepple[1]

[1]Alfred-Wegener-Institut Helmholtz-Zentrum für Polar- und Meeresforschung, Research Unit Potsdam, Telegrafenberg A45, 14473 Potsdam, Germany
[2]University of Potsdam, Institute of Physics and Astronomy, Karl-Liebknecht-Str. 24/25, 14476 Potsdam, Germany

**Correspondence:** Thomas Münch (thomas.muench@awi.de)

**Abstract.** Ice-core-based records of isotopic composition are a proxy for past temperatures and can thus provide information on polar climate variability over a large range of timescales. However, individual isotope records are affected by a multitude of processes that may mask the true temperature variability. The relative magnitude of climate and non-climate contributions is expected to vary as a function of timescale, and thus it is crucial to determine those temporal scales at which the actual signal dominates the noise. At present, there are no reliable estimates of this timescale dependence of the signal-to-noise ratio (SNR). Here, we present a simple method that applies spectral analyses to stable-isotope data from multiple cores to estimate the SNR, and the signal and noise variability, as a function of timescale. The method builds on separating the contributions from a common signal and from local variations and includes a correction for the effects of diffusion and time uncertainty. We apply our approach to firn-core arrays from Dronning Maud Land (DML) in East Antarctica and from the West Antarctic Ice Sheet (WAIS). For DML and decadal to multicentennial timescales, we find an increase of the SNR by nearly one order of magnitude ($\sim 0.2$ at decadal and $\sim 1.0$ at multicentennial scales). The estimated spectrum of climate variability also shows increasing variability towards longer timescales, contrary to what is traditionally inferred from single records in this region. In contrast, the inferred variability spectrum for WAIS stays close to constant over decadal to centennial timescales, and the results even suggest a decrease in SNR over this range of timescales. We speculate that these differences between DML and WAIS are related to differences in the spatial and temporal scales of the isotope signal, highlighting the potentially more homogeneous atmospheric conditions on the Antarctic Plateau in contrast to the marine-influenced conditions on WAIS. In general, our approach provides a methodological basis for separating local proxy variability from coherent climate variations which is applicable to a large set of palaeoclimate records.

## 1 Introduction

Ice cores represent key archives for studying polar climate variability on timescales beyond instrumental observations. The isotopic composition of water stored in the ice serves as a proxy for reconstructing past temperature variations (Dansgaard, 1964; Jouzel and Merlivat, 1984; Jouzel et al., 2003) over a wide range of timescales ranging from subannual to glacial–interglacial variations. This can provide valuable insights into the timescale dependence of climate variability which is thought

to be an inherent property of the climate system (Hasselmann, 1976; North et al., 2011; Lovejoy et al., 2013; Rypdal et al., 2015).

However, the interpretation of isotope records in terms of local atmospheric temperatures is complicated by a multitude of processes that distort the original relationship present in precipitation (e.g., Fujita and Abe, 2006; Sjolte et al., 2011; Steen-Larsen et al., 2011; Touzeau et al., 2016; Casado et al., 2018). To a first approximation, changes in isotopic composition are only recorded in the ice if there is snowfall, while the role of water vapour exchange processes in between precipitation events is still debated (Steen-Larsen et al., 2011; Stenni et al., 2016; Casado et al., 2018; Ritter et al., 2016; Münch et al., 2017a). Both the seasonality and interannual variability in the seasonality of precipitation events is known to strongly affect the isotopic composition of snow layers by introducing a bias to the mean values, e.g., the annual average (Steig et al., 1994; Sime et al., 2009; Laepple et al., 2011), and by adding variability to the signal (Persson et al., 2011; Laepple et al., 2018). Furthermore, uneven deposition of snow, in combination with steady and strong surface winds, lead to constant erosion, drift and vertical mixing of the surface snow, giving rise to strong spatial variability (Fisher et al., 1985; Münch et al., 2016, 2017a). Finally, once the snow is deposited, the diffusion of vapour through the firn column smoothes the isotope variations (Johnsen, 1977; Whillans and Grootes, 1985). This significantly reduces the high-frequency power spectral density of the variations (Johnsen et al., 2000; van der Wel et al., 2015) and substantially shapes the isotope variability (Laepple et al., 2018). Overall, these processes are to a first approximation not directly linked to variations in temperature, and therefore add a significant amount of noise to the isotope records, especially in low-accumulation regions on the Antarctic Plateau. This has been demonstrated by the low representativity of individual ice-core measurements (Karlöf et al., 2006; Münch et al., 2016) and questions the usability of a direct interpretation of high-resolution isotope variations in terms of temperature variability (Laepple et al., 2018), particularly when the climate signal itself is only relatively weak. While deep ice cores show a consistent picture for the strong glacial–interglacial variations (e.g., Jouzel et al., 2007), it may be questionable whether the Holocene variability recorded in individual cores depicts the true temperature variability (Kobashi et al., 2011). Spatial or temporal averaging of isotope records thus provide important tools to reduce overall noise (Fisher et al., 1996; Münch et al., 2016; Stenni et al., 2017).

Previous studies provided first insights into the relationship between climate signal and noise for short spatial and temporal scales (Fisher et al., 1985; Münch et al., 2016). However, an extension to longer scales, which is important for the interpretation of Holocene temperature reconstructions, is still missing. Furthermore, in order to correct the isotope inferred variability estimates for the noise contribution, it is necessary to know the variance of the noise across timescales, i.e. its spectral shape. Here, we present a simple spectral method to separate the local noise from spatially coherent signals using information from several isotope records, including a correction for diffusion and time uncertainty. We apply our model to two spatial arrays of firn cores: (1) from Dronning Maud Land in East Antarctica, and (2) from the West Antarctic Ice Sheet. Our objective is to derive an improved estimate of the temperature variability in both regions and to learn about the timescale dependence of the signal-to-noise ratio in ice-core isotope data. For Dronning Maud Land, our results confirm the noise levels inferred in previous studies on short temporal scales (Münch et al., 2016, 2017a) and also suggest white noise on longer timescales, which results in an increase of the isotopic signal-to-noise ratio (SNR). Unexpectedly, the SNR inferred from the West Antarctic data is found

to show the opposite timescale dependence. These results may point towards marked differences in the spatial and temporal scales of the isotope signals and reveal gaps in our current understanding of the underlying processes.

## 2   Data and methods

### 2.1   Isotope records from Dronning Maud Land and West Antarctica

We analyse published oxygen isotope records of annually dated firn cores from two contrasting Antarctic regions (Table 1): (1) Dronning Maud Land (DML) on the Antarctic Plateau, and (2) the West Antarctic Ice Sheet (WAIS). While the DML core sites are located in a rather flat region relatively isolated from the coast (average elevation 2900 m), the WAIS core sites are lower in elevation (1600 m on average) and therefore potentially more influenced by marine conditions.

For DML, we use a total of 15 records which were collected during the EPICA (European Project for Ice Coring in Antarc-
tica) pre-site survey (Oerter et al., 2000) and published in Graf et al. (2002a). All records cover at least the last $\sim 200$ yr and form our data set DML1. Three of these records (B31–B33) span the last $\sim 1000$ yr and are therefore included in a second separate data set for this time span (DML2). Core B32 is in close proximity ($\sim 1$ km) to the deep EPICA DML (EDML) ice-core site at Kohnen Station (EPICA community members, 2006). The record chronologies were established from seasonal layer counting of chemical impurity records constrained by tie points from the dating of volcanic ash layers (Graf et al., 2002a). The
resulting uncertainty of the chronologies has been reported to be $\sim 2$ % of the time interval to the nearest tie point (Graf et al., 2002a), which translates to a maximum time uncertainty of $\sim 1.2$ yr for the short and of $\sim 3.5$ yr for the long records. Since our method (Sect. 2.2) relies on all records having equal lengths, we restrict the time span for the DML1 data set to the period from 1801–1994 CE and to the period from 1000–1994 CE for the DML2 data set.

The WAIS data set selected for this study consists of five isotope records (Steig et al., 2013) collected during the US ITASE
(International Trans-Antarctic Scientific Expedition) project (Mayewski et al., 2005), including the core WDC2005A from the WAIS Divide ice core (WDC; WAIS Divide Project Members, 2013), and cover the time interval from 1800–2000 CE. The oxygen isotope data of cores ITASE-2000-4 and ITASE-2000-5, previously published at subannual resolution (Steig et al., 2013), has been block-averaged in this study to obtain annual resolution data. The core selection has been based on the constraint to cover a similarly large area and sufficiently long time period to allow a meaningful comparison with the
DML1 records. The relative time uncertainty between the WAIS records, based on dating through annual layer counting of chemical trace species validated by identification of volcanic marker horizons, was reported to be no more than 1 yr (Steig et al., 2005).

### 2.2   Model for the separation of signal and noise in the spectral domain

Our approach in general assumes that individual isotope records from a certain region contain two contributions: (i) a signal
common to all cores from that region, and (ii) independent noise components which are, for example, related to spatial variability from redistribution of snow (stratigraphic noise) or to precipitation intermittency. By utilising several records we can

**Table 1.** Overview of the firn cores (sorted into three data sets) used in this study. Listed are the covered time span of each core array (in yr CE), the number of records in each array ($n$), the region of origin (latitude/longitude range), the range of site elevations, local accumulation rates ($\dot{b}$) and 10 m firn temperatures ($T_{\text{firn}}$) as reported in the original publications, and the range of intercore distances ($d$). The range of WAIS firn temperatures is based on ERA-Interim annual mean anomalies (Dee et al., 2011) with respect to the WDC2005A site.

| Core array (Time span) | $n$ | Region (Lat./Lon.) | Elevation m a.s.l. | $\dot{b}$ $\text{kg}\,\text{m}^{-2}\,\text{yr}^{-1}$ | $T_{\text{firn}}$ °C min | $T_{\text{firn}}$ °C max | $d$ km | Data reference |
|---|---|---|---|---|---|---|---|---|
| DML1[a] (1801–1994) | 15 | 74.5–75.6° S 6.5° W–6.5° E | 2630–3160[d] | 40–90[d] | −46[d] | −40[d] | 1–370 | Graf et al. (2002b) |
| DML2[b] (1000–1994) | 3 | 75.0–75.6° S 3.4° W–6.5° E | 2670–3160[d] | 50–60[d] | −46[d] | −44[d] | 120–280 | Graf et al. (2002b) |
| WAIS[c] (1800–2000) | 5 | 77.7–80.6° S 111.2–124.0° W | 1350–1830[e,f] | 140–220[e,f] | −30[f,g] | −28[f,g] | 20–340 | Steig (2013) |

[a]Firn cores FB9804, FB9805, FB9807–FB9811, FB9813–FB9817, B31–B33 [b]Firn cores B31–B33 [c]Firn cores WDC2005A, ITASE-1999-1, ITASE-2000-1, ITASE-2000-4, ITASE-2000-5 [d]Oerter et al. (2000) [e]Kaspari et al. (2004) [f]WAIS Divide Project Members (2013) [g]Dee et al. (2011)

disentangle both contributions and estimate the underlying common and noise signals. The approach is similar to the analysis of variance (e.g., Fisher et al., 1985) but uses the power spectra of the time series to derive timescale-dependent estimates.

More formally, given a core array of $n$ isotope records, the power spectral density (PSD) of an individual record from site $i$, $\mathcal{X}_i(f)$, where $f$ denotes frequency, is the sum $\mathcal{X}_i(f) = \mathcal{C}(f) + \mathcal{N}_i(f)$, where $\mathcal{C}(f)$ and $\mathcal{N}_i(f)$ are the original spectra of the common signal and the noise component, respectively, prior to the smoothing by molecular diffusion of water vapour within the porous firn (Johnsen et al., 2000; van der Wel et al., 2015). To relate this with the actually measured signal $\hat{\mathcal{X}}_i(f)$, we additionally have to account for the measurement process which adds additional noise to the diffused record. $\hat{\mathcal{X}}_i(f)$ is thus given by

$$\hat{\mathcal{X}}_i(f) = \mathcal{G}_i(f)\left[\mathcal{C}(f) + \mathcal{N}_i(f)\right] + \Sigma. \tag{1}$$

Here, $\mathcal{G}_i(f)$ is a linear transfer function that acts as a low-pass filter and accounts for the diffusion process (Appendix A), and $\Sigma$ is the measurement error. At annual resolution, $\Sigma$ is typically at least one order of magnitude smaller than the stratigraphic noise level (Münch et al., 2016) and is therefore neglected in the following. We now calculate the mean spectrum, $\mathcal{M}(f)$, of all individual spectra $\hat{\mathcal{X}}_i(f)$. Assuming that the statistical properties of the individual noise terms are identical for all $n$ records, we obtain

$$\mathcal{M}(f) = \frac{1}{n}\sum_{i=1}^{n}\hat{\mathcal{X}}_i(f) = \overline{\mathcal{G}}(f)\left[\mathcal{C}(f) + \mathcal{N}(f)\right], \tag{2}$$

where $\overline{\mathcal{G}}(f) = 1/n\sum_{i=1}^{n}\mathcal{G}_i(f)$ is the average diffusion transfer function. In contrast to forming the spectral mean, we can also average the $n$ isotope records in the time domain and then calculate the PSD of this "stacked" record. Here, the noise component

will be reduced by a factor of $n$ compared to the mean spectrum in (2) since we assume independent noise between the sites. Additionally, we have to take into account the time uncertainty of the dated records. This does not affect the overall shape of broadband spectra (Rhines and Huybers, 2011) but diminishes the correlation between the records (Haam and Huybers, 2010) and thus their spectral coherence, which reduces the variance of the common signal in the stacked record. The PSD of the stacked record is thus

$$\mathcal{S}(f) \approx \overline{\mathcal{G}}(f) \left[ \Phi(f)\mathcal{C}(f) + \frac{1}{n}\mathcal{N}(f) \right], \tag{3}$$

where $\Phi(f)$ is a linear transfer function accounting for the effect of time uncertainty. Applying the average diffusion transfer function, $\overline{\mathcal{G}}(f)$, also to the spectrum of the stacked record is a valid approximation in the case of similar $\mathcal{G}_i(f)$ which we confirmed for our records (Appendix A). From the expressions for the mean spectrum ($\mathcal{M}$, Eq. 2) and the spectrum of the stacked record ($\mathcal{S}$, Eq. 3) we can now derive expressions for the spectra of the common signal $\mathcal{C}$ and the noise $\mathcal{N}$ (omitting the explicit frequency dependence in the notation here),

$$\mathcal{C} = \frac{n}{n - \Phi^{-1}} \Phi^{-1} \overline{\mathcal{G}}^{-1} \left[ \mathcal{S} - n^{-1}\mathcal{M} \right], \tag{4a}$$

$$\mathcal{N} = \frac{n}{n - \Phi^{-1}} \overline{\mathcal{G}}^{-1} \left[ \mathcal{M} - \Phi^{-1}\mathcal{S} \right]. \tag{4b}$$

### 2.3 Transfer functions for vapour diffusion and time uncertainty

For estimating the transfer functions for diffusion and time uncertainty, whose inverses are used to correct the signal and noise spectra (Eqs. 4), we use numerical simulations since no closed form expressions are available. For this, we create surrogate records mimicking the individual core arrays and simulate the effects of diffusion and time uncertainty on the surrogate spectra of the stacked records. We model the firn diffusion as the convolution of the original time series with a Gaussian kernel (Johnsen et al., 2000). The width of the kernel is set by the diffusion length $\sigma$, which is site-specific and a function of depth. For modelling the time uncertainty we use the approach of Comboul et al. (2014). Model parameters are the rates of missing and double-counted annual layers which we set to match the reported time uncertainties of the isotope records. Appendix B gives a detailed description of the simulations including the estimated transfer functions (Fig. B1). Because the diffusion correction ($\overline{\mathcal{G}}^{-1}$ in Eqs. 4) strongly amplifies both the raw spectra as well as their uncertainties on the fast frequencies, we restrict our analyses to the frequency region where the estimated transfer function $\overline{\mathcal{G}} \geq 0.5$, equivalent to a correction factor $\leq 2$ in Eqs. 4. This avoids large uncertainties and translates to a maximum frequency that is used for the analyses (hereafter: cutoff frequency) of $1/5\,\mathrm{yr}^{-1}$ for DML1, $1/8.5\,\mathrm{yr}^{-1}$ for DML2 and $1/2.8\,\mathrm{yr}^{-1}$ for WAIS (Fig. B1).

### 2.4 Timescale-dependent estimate of the signal-to-noise ratio

The frequency-dependent SNR, $F(f)$, is defined as the ratio of the signal and noise spectra,

$$F(f) = \frac{\mathcal{C}(f)}{\mathcal{N}(f)}. \tag{5}$$

As we explicitly neglect measurement noise, this quantity is unaffected by the effect of diffusion or its correction (Eqs. 4) but directly influenced by time uncertainty which biases the uncorrected SNR towards zero. Typically, firn or ice-core isotope records are averaged onto a fixed temporal resolution $\Delta t$ (the "averaging period") during preprocessing. The signal-to-noise variance ratio after this averaging step is given by

$$\overline{F}(f_{\mathrm{Nyq}}) = \frac{\int_{f_0}^{f_{\mathrm{Nyq}}} \mathcal{C}(f)\,\mathrm{d}f}{\int_{f_0}^{f_{\mathrm{Nyq}}} \mathcal{N}(f)\,\mathrm{d}f}, \tag{6}$$

where $f_0$ is the lowest frequency of the spectral estimates and $f_{\mathrm{Nyq}}$ the Nyquist frequency for the chosen averaging period, i.e. $1/(2\Delta t)$. Since our records have different lengths, we choose a common minimum value for $f_0$ of $1/100\,\mathrm{yr}^{-1}$ for the integrations in (6). The value of $\overline{F}$ can then be used to obtain the correlation between the time series of the common signal $c$ and a stacked record $\overline{x}$ built from the spatial average of $n$ individual records (Fisher et al., 1985):

$$r(c,\overline{x})(f_{\mathrm{Nyq}}) = \frac{1}{\sqrt{1 + \left(n\overline{F}(f_{\mathrm{Nyq}})\right)^{-1}}}. \tag{7}$$

## 2.5   Spectral analysis

Missing years in the published records ($\sim 1.6\,\%$ of all data points) are linearly interpolated. Power spectra are then estimated using Thomson's multitaper method with three windows (Percival and Walden, 1993) with linear detrending before analysis. This approach is known to introduce a small bias at the lowest frequencies, and we omit the lowest frequency from all plots and calculations. In general, spectral smoothing is necessary to improve the quality of the estimates from short time series. Here, we use a Gaussian smoothing kernel with varying width proportional to the applied frequency and thus constant in logarithmic frequency space (Kirchner, 2005). The smoothing width in logarithmic units is chosen to be $0.1$ for the WAIS data and $0.15$ for the DML data. To avoid biased estimates at the low- and high-frequency boundaries, the kernel is truncated on both sides to maintain its symmetry. Logarithmic smoothing preserves power-law scaling of spectra and is thus useful for climate spectra.

## 3   Results

### 3.1   Illustrating the methodological approach

In order to illustrate our method (Sect. 2.2), we first use the DML1 data set to demonstrate the individual steps involved in the analysis.

Each power spectrum derived from an individual record of the DML1 firn-core array provides a timescale-dependent representation of the isotope variations in the study region (Fig. 1, thin grey lines). The differences between the individual spectra are not only due to differences in the isotope variability between the records, but also caused by the spectral uncertainty from the finite length of each record. The mean spectrum, $\mathcal{M}$, of all individual spectra reduces this spectral uncertainty and thus provides a more robust estimate of the single records' spectrum (Fig. 1, black line). It shows a two-part structure with a near constant PSD above decadal timescales (i.e. is "white") and a strong decrease in spectral power towards shorter timescales which is expected from vapour diffusion in the firn (Eq. 2 and Appendix B).

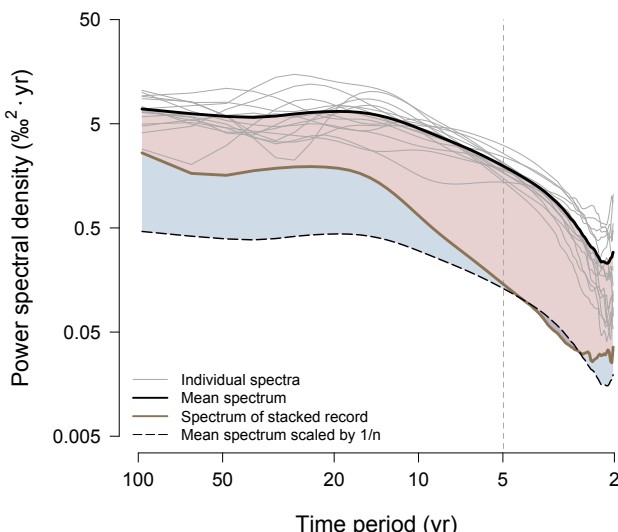

**Figure 1.** Detailed results of estimated PSD for the DML1 data set. Thin grey lines show the individual power spectra for each record with the mean spectrum indicated by the thick black line. The dashed black line shows the null hypothesis according to which all isotope variations are noise; the brown line depicts the spectrum from averaging all records in the time domain (the "stacked" record). The extent of the blue (red) shadings is proportional to the uncorrected PSD of the signal (noise) (Eqs. 4). The vertical dashed line denotes the cutoff period for the diffusion correction (see Methods).

The mean spectrum divided by the number of records ($\mathcal{M}/n$; here, $n=15$; Fig. 1, dashed line) is the expected spectrum if all isotope variations present in the firn-core records were independent noise. In comparison, averaging in the time domain across records that contain noise and additionally a common (i.e. spatially coherent) signal, will result in a spectrum $\mathcal{S}$, where the noise component is also reduced by $1/n$ but with the common signal left unaltered (Eq. 3, neglecting time uncertainty), and which is thus located between the mean spectrum and the mean spectrum divided by $n$. This spectrum $\mathcal{S}$ for the DML1 stack (Fig. 1, brown line) is for short timescales (periods from $\sim 3$ to $5\,\mathrm{yr}$) nearly identical to the mean spectrum divided by $n$, which suggests that here the variability of the individual records is consistent with the null hypothesis of independent noise. The divergence from the white-noise level close to the 2-yr Nyquist period is likely an artefact from the jitter of the annual averages as a result of the uncertainty in defining annual depth increments upon dating. In contrast to the short periods below $10\,\mathrm{yr}$, the individual records clearly contain a common isotope signal on longer timescales that "survives" the averaging process when building the stacked record and which increases in power with increasing timescale (Fig. 1, brown line). Hence, the differences between $\mathcal{S}$ and $\mathcal{M}/n$ and between $\mathcal{S}$ and $\mathcal{M}$, respectively, inform us about the average signal and noise content of the individual isotope records, but they need to be corrected for the residual amount of noise contained in $\mathcal{S}$ and for the loss of variance from diffusion and time uncertainty (Eqs. 4).

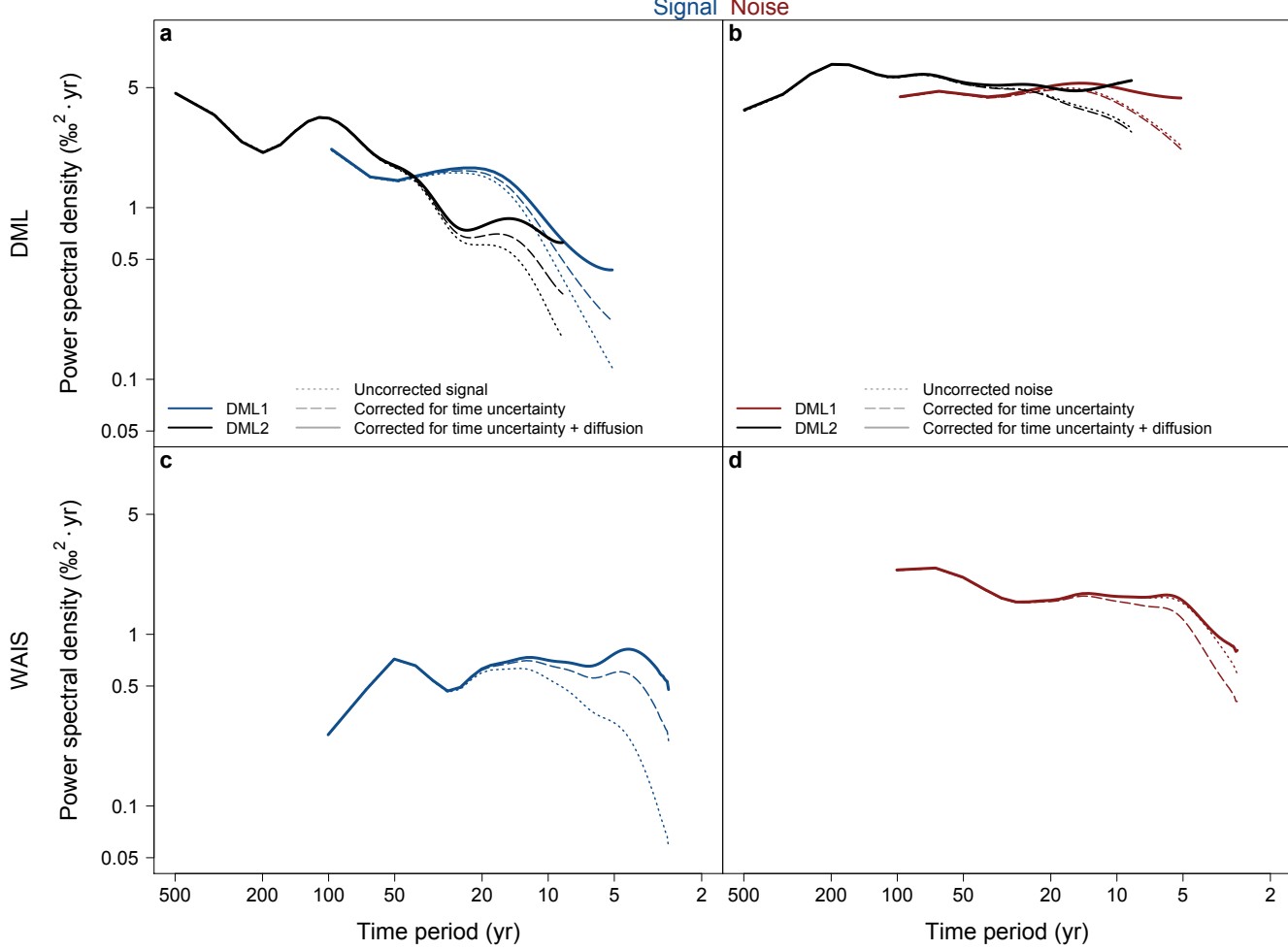

**Figure 2.** Estimated signal (left) and noise (right) spectra of Antarctic isotope records. Results are shown based on the spectral correction model for the records from Dronning Maud Land (DML, top row) and from the West Antarctic Ice Sheet (WAIS, bottom row). The raw estimates (dotted lines) show the spectra prior to any correction, while corrected estimates differentiate between the correction for time uncertainty only (dashed lines) and the full correction including time uncertainty and diffusion (solid lines). For DML, results include both the short, 200 yr long records (DML1; blue lines in **a**, red lines in **b**) and the longer records covering the last 1000 yr (DML2; black lines).

## 3.2 Timescale dependence of DML and WAIS signal and noise variability

After this detailed description for DML1, we now turn towards the results of the estimated signal and noise spectra for all three data sets. In general, the shape of the signal spectra is, as a result of the corrections, clearly distinct from the mean spectrum of the individual isotope variations. This is seen, for example, in the corrected DML1 signal spectrum which indicates a much more steady increase in PSD from short to long timescales (Fig. 2a, solid blue line) as compared to the mean spectrum (Fig. 1, black line). This is confirmed by the three 1000 yr long DML2 records whose signal exhibits a roughly similar power spectrum in the range of timescales that overlap ($\sim$ 10 to 100 yr period). We partly expect this since the longer cores are also part of the DML1 data set, but the increase in signal power in this frequency band seems to be a general feature over the entire last 1000 yr. In addition, the DML2 signal spectrum shows a further and similar increase for timescales beyond the 100 yr period. This change in spectral shape from the mean to the signal spectrum results from two contributions in the correction process: Firstly, the average isotope variability is corrected for the noise contribution ("uncorrected signal", dotted lines in Fig. 2a). This correction is mostly apparent on the longer timescales leading to a higher slope in PSD of the signal spectrum compared to the mean spectrum (for DML1, the signal increases from the 10 to 100 yr period by a factor of $\sim$ 2.6, the mean spectrum by a factor of $\sim$ 1.4). Secondly, the corrections for the loss in spectral power by the effects of diffusion and time uncertainty (here important for time periods from $\sim$ 20 to 5 yr) lead to a smaller increase in PSD of the signal spectra with increasing timescale as compared to the raw estimates (compare dotted, dashed and solid lines in Fig. 2a). In sharp contrast to DML, the corrections for the WAIS records yield a signal spectrum (Fig. 2c, solid line) that exhibits a rather flat appearance throughout and indicates decreasing PSD towards centennial timescales. Much of the flat character is caused by the diffusion and time uncertainty corrections which strongly amplify the raw signal power on the subdecadal timescales; the decrease in PSD on the long timescales is a result of the correction for the noise contribution. We note, however, that the spectral uncertainty at the lowest frequency is high since fewer data points contribute to the estimated PSD for lower frequencies when using logarithmic smoothing.

A difference between both regions can also be seen in the noise spectra (Fig. 2b+d). Prior to any correction, the DML1 and DML2 noise spectra are different on shorter timescales ($\lesssim$ 20 yr period), but become consistent to each other after the correction showing essentially white PSD (Fig. 2b). In comparison, the corrected WAIS noise spectrum shows an increase of spatially incoherent isotope variations towards longer timescales, which correlates with the decrease in signal power on centennial timescales (Fig. 2c).

The corrected signal and noise spectra directly provide an estimate of the timescale dependence of the SNR (Eq. 5). To derive a single estimate for DML, we combine the DML2 spectra for timescales above the decadal period with the respective DML1 spectra from the subdecadal frequency band. Again, the results are very different between DML and WAIS (Fig. 3). For DML, the SNR shows a continuous increase with timescale with values ranging from $\sim$ 0.1 at the 5 yr period to $\lesssim$ 0.2 at the decadal timescale and increasing further to $\sim$ 1 at multicentennial timescales. The estimate for WAIS exhibits the opposite timescale dependence with SNR values of $\sim$ 0.5–0.7 for interannual timescales that continuously decrease to $\sim$ 0.1 at centennial timescales.

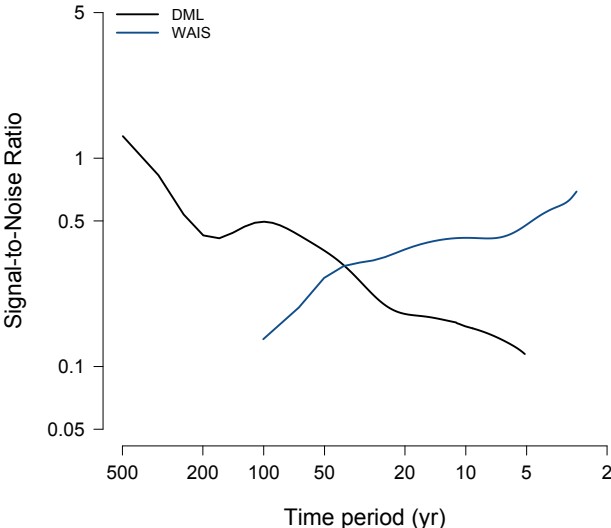

**Figure 3.** Estimated timescale dependence of ice-core isotope signal-to-noise ratios. Results are shown for the DML (black) and WAIS (blue) isotope records. The results for DML are based on combining the spectra from DML1 and DML2 (see text).

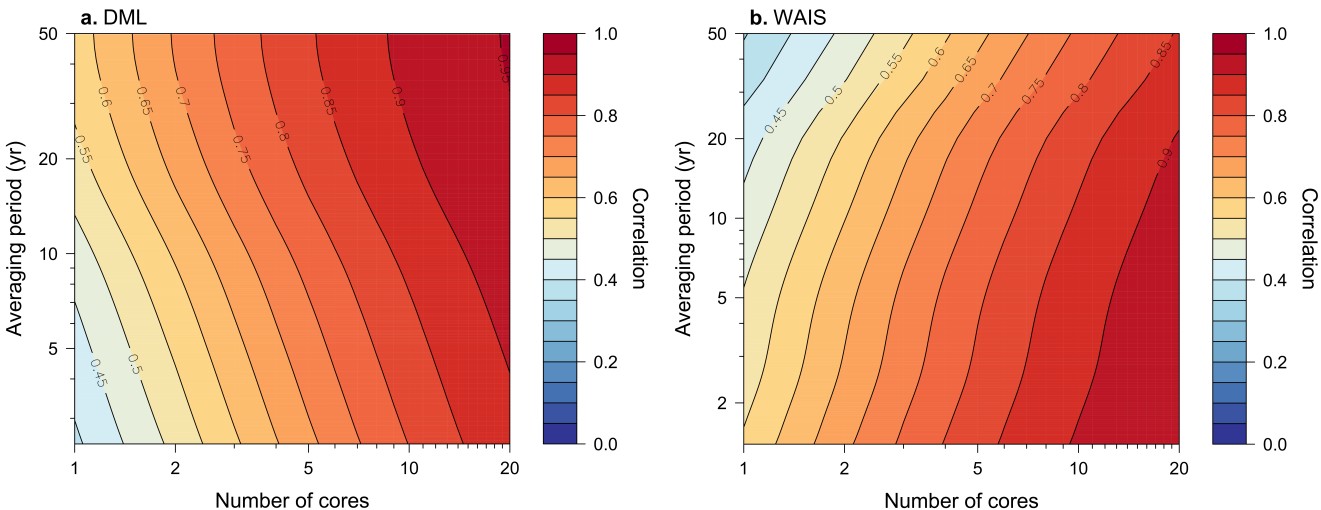

**Figure 4.** Estimated correlation of stacked isotope records from (**a**) DML and (**b**) WAIS with their common signal as a function of the averaging period and the number of firn cores included in the average. The correlations are based on Eq. (7) with the same signal and noise spectra that were used for Fig. 3.

A complementary picture is obtained from the expected correlation between the time series of a stacked record and the underlying common signal as a function of both the number of averaged records and the temporal averaging period (Eq. 7,

Fig. 4). For DML, the correlation for an individual record at interannual resolution is rather low with roughly 0.4–0.5 (Fig. 4a); for WAIS, this correlation is slightly higher ($\sim$ 0.5–0.6, Fig. 4b). For longer averaging periods, the correlation for DML shows a steady increase with the averaging period, in line with the increase in SNR, reaching values around 0.6 for single records and multidecadal averaging periods (Fig. 4a). Naturally, the correlation further increases with the number of averaged cores. For WAIS, the correlation with the underlying signal decreases with the averaging period and only increases by averaging more records (Fig. 4b).

## 4 Discussion

### 4.1 Physical interpretation of the analysis

We presented a method and the results of separating the variability recorded by Antarctic isotope records into two contributions: local variations ("noise", Eq. 4b) and spatially coherent variations ("signal", Eq. 4a). We now assess the physical meaning of these terms by discussing the relevant spatial scales of the major processes that influence the isotopic composition of the records: (i) temperature variations, (ii) atmospheric circulation, (iii) precipitation intermittency, and (iv) stratigraphic noise.

Classically, the isotopic composition of firn and ice cores is interpreted as being related to variations in local air temperature (Dansgaard, 1964; Jouzel and Merlivat, 1984; Jouzel et al., 2003). The extent to which spatially distributed isotope records are influenced by a common temperature signal then depends on the decorrelation scale of the temperature anomalies, which is, for annual mean temperatures, typically of the order of hundreds to thousands of kilometres and increases with timescale (Jones et al., 1997). For our study regions, annual mean temperature variations from the ERA-Interim reanalysis (Dee et al., 2011) exhibit decorrelation scales of $\sim 1200\,\text{km}$ (Appendix C) which are much larger than the individual intercore distances ($< 400\,\text{km}$, Table 1). This implies that, if the temperature variations were fully recorded by the array of firn-core records, they would to a large extent contribute to the estimated signal spectrum. In fact, for the temperature reanalysis fields in the study regions, the average of all correlations between sites separated by less than the maximum intercore distances (Fig. C1) suggests that the estimated signal spectra for DML (WAIS) would contain $87\,\%$ ($94\,\%$) of the total temperature variability while the remaining fraction would be interpreted as noise.

However, changes in large-scale atmospheric circulation can lead to variations in the source and the pathways of the moisture, which can affect the isotopic composition of the precipitation that is formed, independent of local temperature changes (e.g., Schlosser et al., 2004). In general, the spatial scales of such variations are unclear. For the studied DML core array, which is located on the rather flat and remote Antarctic Plateau, one could expect that the effect is small and possibly spatially coherent, thus contributing to the estimated signal term. For WAIS, isotope data have been interpreted to also reflect changes in atmospheric circulation and sea-ice cover of the adjacent oceans (Küttel et al., 2012; Steig et al., 2013). These processes might exhibit, through the ice-sheet topography, a regional expression at the individual firn-core sites, which are generally lower in elevation and located near the ice divide, as suggested by the significant spatial differences in observed intercore correlations (Küttel et al., 2012). Such effects would affect the estimated noise component.

Major additional contributions to the overall variability of isotope data arise from precipitation intermittency, i.e. interannual variations in the seasonality of precipitation events (Persson et al., 2011; Laepple et al., 2018), and from stratigraphic noise (Fisher et al., 1985; Karlöf et al., 2006; Münch et al., 2016) created during deposition of the snow. Both processes exhibit significantly different decorrelation scales. The decorrelation scale of precipitation intermittency is related to the decorrelation scale of the precipitation itself, which is expected to be coherent on a local scale (i.e. $\sim 100\,\mathrm{m}$) and to decorrelate on scales generally smaller than the temperature decorrelation scales. In both regions, the similar analyses of ERA-Interim data as for the temperature variations suggests decorrelation scales of the precipitation of $300$–$500\,\mathrm{km}$ (Appendix C). This is supported by an analysis of measurements from autonomic weather stations (Reijmer and van den Broeke, 2003) that indicate that accumulation in DML arises from many small ($1$–$2\,\mathrm{cm}$) and few larger events ($\gtrsim 5\,\mathrm{cm}$) of snowfall, where the major events occur only a few times per year without any clear seasonality but often over large areas. In contrast, stratigraphic noise is a short-scale phenomenon. Its generation is related to the uneven deposition (Fisher et al., 1985) and the constant erosion and redeposition of the surface snow by wind (Münch et al., 2016). Both processes are connected to the local surface undulations as these directly interact with the snow deposition but also strongly shape the near-surface wind field. This is suggested by the observed decorrelation scale of the stratigraphic noise at EDML ($< 5\,\mathrm{m}$, Münch et al., 2016) which is similar to the typical scale of the surface undulations.

These differences in decorrelation scales also become apparent when analysing the relative contributions of both processes to the total isotope variability. At EDML, stratigraphic noise provides $\sim 50\,\%$ of the total variance at the seasonal timescale, as suggested from the average correlation of individual shallow isotope profiles separated above ($> 5\,\mathrm{m}$) the decorrelation scale of the stratigraphic noise but below $1\,\mathrm{km}$ (Münch et al., 2016, 2017a). A much higher noise level of at least $80\,\%$ of the total variance (i.e. higher by a factor of $\geq 1.6$) has been independently inferred from comparing the observed seasonal isotope variability to the expectation from a profile that is a mixture of a deterministic seasonal cycle and diffused noise (Laepple et al., 2018). The apparent mismatch with the noise level from stratigraphic noise can be reconciled by asserting that a significant part of the common isotope signal at EDML on the local scale is coherent noise from precipitation intermittency, leading to an underestimation of the total noise level when analysing only the interprofile correlations. On the larger spatial scales of the here analysed firn-core arrays ("array scale"), the effect of precipitation intermittency should then appear, at least partly, as a noise term since the spatial precipitation patterns are expected to decorrelate on these scales. Therefore, in general, the variability contribution by stratigraphic noise will fully appear in the noise spectra, while precipitation intermittency is expected to partly contribute to the noise spectra but also partly appear in the signal spectra.

In summary, given the large decorrelation scales of atmospheric temperature variations and the generally smaller scales of the other terms, one could interpret the estimated isotope signal spectra to a first approximation as temperature signals. However, this clearly will be an upper bound of the true temperature signal since also other processes can lead to spatially coherent isotope signals. Furthermore, we have neglected the transfer function from isotopic ratios to temperatures, and other less constrained effects that affect the isotope signal from the atmosphere to the snow (e.g., Casado et al., 2018).

## 4.2 Interpretation of the estimated signal and noise spectra

The raw noise spectra, i.e. prior to correction, derived from the two DML data sets exhibit a clear imprint from the diffusional smoothing in the firn. This is suggested by their common decrease in PSD towards shorter time periods (Fig. 2b), since diffusion acts stronger on higher frequencies. It is corroborated by comparing the loss in PSD between the two data sets, which for DML2 is stronger towards the high-frequency end and also extends further towards longer time periods. This is due to the stronger diffusional smoothing in the older sections of the cores that are only contained in the DML2 records, since the diffusion process had more time to act there since deposition of the snow. The applied correction reconciles both noise estimates, as it takes these differences into account, and reveals a close-to-constant noise level ("white noise") across the range from subdecadal to multicentennial timescales.

This near constancy of the noise level suggests that the noise-creating processes are independent of the timescale. This seems plausible for stratigraphic noise as it is also indicated by the observed small memory in the interannual variations of the surface topography at EDML (Laepple et al., 2016; Münch et al., 2016). Furthermore, we would not expect strong changes in surface wind regimes or depositional characteristics for the rather stable climatic conditions over the studied time periods. To test whether the effect of precipitation intermittency additionally contributes to the noise spectrum, as suggested from the decorrelation scales of the ERA-Interim precipitation fields, we apply our spectral analysis to the available isotope profiles on the local scale ($\sim 100\,\mathrm{m}$) at EDML (Münch et al., 2017a, b) and compare the estimated noise spectra between the local scale and the array scale (Fig. 5). Although the spectral estimates do not overlap, the results indicate an offset between the noise levels at both spatial scales. At the 5 yr period, the PSD of the noise on the array scale is about 1.7 times higher than at the local scale. Some increase in the noise level is expected from the small contribution of uncorrelated temperature variability, but the $\sim 1.7$-fold increase suggests that most of the additional noise arises from spatially incoherent precipitation intermittency.

This supports the interpration of the estimated DML signal spectra (Fig. 2a) as temperature variability, assuming that the influence of atmospheric circulation changes is small. Fitting a power law of the form $f^{-\beta}$, where $f$ denotes frequency, yields a slope of the DML2 signal spectrum of roughly $\beta \sim 0.6$ for variations above the 20 yr period. This is higher than the slopes of the mean spectra of the single records ($\beta \sim 0.2$ for DML2, $\beta \sim 0$ for DML1) which underlines again that the high noise level in individual records strongly masks the true spectral shape of the signal. Such a power-law increase in temperature variability is not unexpected since it was also observed in spectra from marine-proxy-inferred sea surface temperature variations (Laepple and Huybers, 2014) and from other proxy and instrumental temperature records (e.g., Pelletier, 1998; Huybers and Curry, 2006).

The resulting SNR (Fig. 3) of the DML data illustrates at which timescales the signal dominates the isotope variability recorded in individual records, and how this translates into the representativity of isotope variations when averaging records in space and time (Fig. 4). We can test the validity of our spectral approach by comparing the estimated SNR to published estimates. Graf et al. (2002a) analysed the same data set as the present study and found a SNR at annual resolution of $\overline{F} = 0.14$ based on the average intercore correlations ($r = 0.35$). Similar values have been obtained using a larger set of cores from the same region that cover a shorter time period (Altnau et al., 2015). From integrating the estimated signal and noise spectra up

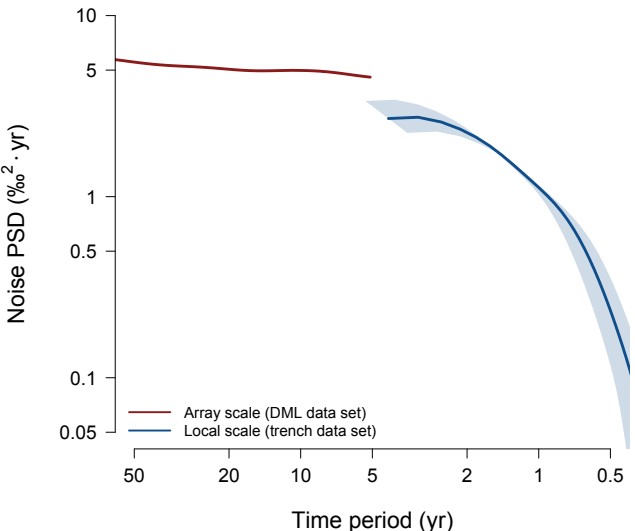

**Figure 5.** Comparison of DML noise spectra as a function of intersite distance. The red curve (array scale, $\sim 100\,\mathrm{km}$) shows the section for periods $\leq 50\,\mathrm{yr}$ of the composite noise spectrum from the DML1 and DML2 firn-core records (Fig. 2b and text). The blue curve (local scale, $\sim 100\,\mathrm{m}$) depicts the noise spectrum as estimated from 22 shallow ($\sim 3.5\,\mathrm{m}$ depth) trench profiles from EDML (Münch et al., 2017a, b). The depth scale of the trench data has been converted into time units using a constant accumulation rate of $25\,\mathrm{cm}$ of snow per year, neglecting the small compression by densification (Münch et al., 2017a). Blue shadings denote the spectral range from an assumed $20\,\%$ uncertainty of the accumulation rate. Note that the trench noise spectrum is not corrected for the diffusional smoothing whose effect is negligible for the trench records at the $5\,\mathrm{yr}$ period and thus does not affect the comparison of the noise levels.

to the minimum averaging period constrained by the diffusion correction ($\sim 2.5\,\mathrm{yr}$) we find a SNR of $\overline{F} \sim 0.2$ (correlation $r \sim 0.4$, Fig. 4a), which is, as expected, slightly higher than the published estimates, since we corrected for the effect of time uncertainty, as well as due to the effect of the slightly different underlying averaging periods. Our results further explain the strong agreement between individual ice cores on glacial–interglacial timescales. Averaging to multidecadal resolution

5 results in a correlation between a single core and the signal of around $0.6$ (Fig. 4a), which rises to $\sim 0.7$ for the available centennial averaging periods. Moreover, the steady increase in SNR on the analysed timescales (Fig. 3) might indicate a further increase in SNR towards even longer timescales. However, clearly our results also underline again that for assessing Holocene temperature variability on timescales shorter than multidecadal, the averaging of records is essential to increase their representativity (Fig. 4a).

10 For WAIS, the higher SNR at interannual timescales as compared to DML (average SNR between 5 and 10 yr periods of $0.43$ for WAIS compared to $0.13$ for DML; Fig. 3) is consistent with the general notion that higher accumulation rates (on average, $\sim 3$ times as high as at DML; Table 1) result in higher SNR (Fisher et al., 1985; Steen-Larsen et al., 2011). However, one would expect, to a first approximation, a similar increase in SNR with timescale in both regions. In contrast to this expectation, the WAIS results show strongly different timescale dependencies with the tendency of a decrease in signal power and an increase

in noise level on longer timescales (Fig. 2), resulting in an overall decrease in SNR (Fig. 3). This explains the only slightly higher correlations at interannual timescales of a stacked isotope record with the common signal at WAIS as compared to DML (Fig. 4), since the correlation is based on the integrated value of the SNR (Eqs. 6 and 7). If they are correct, these findings suggest that there is, in general, no simple scaling of the SNR with accumulation rate but that additional effects need to be involved.

The shape of the signal and noise spectra at subdecadal timescales is sensitive to the diffusion and time uncertainty corrections (Fig. 2), which could thus contribute to the discrepant SNR estimates. While the diffusion correction for DML led to a consistent white-noise level of both noise estimates (Fig. 2b) at these timescales, lending support to our approach, the WAIS noise spectrum keeps decreasing towards higher frequencies even after applying the diffusion correction (Fig. 2d). This result could be caused either by a too weak diffusion correction, or by an overestimation of the time uncertainty leading to an excessive reduction in high-frequency noise levels that cannot be compensated by the low diffusion correction. To test both hypotheses, we varied the strengths of the diffusion and time uncertainty corrections. This has different impacts on the estimated spectra: while the diffusion correction equally applies to both raw signal and noise spectra, the time uncertainty correction has a proportional influence on the signal spectrum but only an indirect influence on the noise spectrum (Eqs. 4). Indeed, halving the time uncertainty would increase the noise spectrum only by a maximum factor of $\sim 1.4$ close to the cutoff frequency, and this therefore cannot reconcile the remaining decrease in the corrected noise spectrum. By contrast, an overall doubling of the diffusion length for all WAIS records would amplify the noise spectrum much stronger with a maximum factor of $\sim 4$ at the cutoff frequency, leading to interannual noise levels similar to those observed for centennial periods. However, such a strong diffusional smoothing at WAIS compared to the expectation seems implausible given that the same physical mechanisms of the diffusion process should be valid for East as well as West Antarctica, and no anomalously high diffusional smoothing has been observed for the upper part of the WAIS Divide ice core (Jones et al., 2017). Additionally, a much stronger diffusion correction would also imply a much steeper decrease of the signal spectrum towards longer timescales (Fig. 2c), contrary to the expectation. We conclude that there is no obvious reason for the applied correction approach to strongly over- or underestimate the WAIS signal and noise spectra on the interannual timescales. Moreover, taking into account also the shape of the sprectra which we find for the signal and noise on the longer timescales (periods from $\sim 30$–$100$ yr), our results might therefore indicate a true increase in local variability at the WAIS sites across the timescales studied and a close-to-constant, or even decreasing, coherent signal variability.

These results suggest firstly that an additional noise process, apart from stratigraphic noise and precipitation intermittency, must contribute to the noise spectrum towards longer timescales. Secondly, the shape of the signal spectrum either implies a nearly white-noise temperature signal, or some process that destroys the coherence of the large-scale temperature field on longer timescales upon recording by the firn-core isotope records. Since there is no obvious reason for a fundamentally different Holocene climate variability in West compared to East Antarctica, the second possibility seems more likely.

WAIS isotope data have been reported to covary with local temperatures, but also with the large-scale atmospheric circulation and the sea-ice cover of the adjacent oceans (Noone and Simmonds, 2004; Küttel et al., 2012; Steig et al., 2013). A pronounced regional imprint on the recorded isotope variability is also suggested from the spatial differences in intercore correlations for

an extended set of the available WAIS firn cores (Küttel et al., 2012). Especially cores east of the WAIS Divide display strong differences in the recorded isotope variability despite their rather small spatial separation (Gregory and Noone, 2008), which is suggested to be related to slow variations in the dominant circulation patterns. While such differences have not explicitly been reported for the core sites west of the divide studied here, we hypothesise that signatures of sea-ice variations or meridional inflow could affect the isotopic composition at the individual firn-core sites to different extents. This is motivated by model-based results (Noone and Simmonds, 2004) linking an increase in sea-ice cover to an earlier ascent of long-range transported air masses to the continent, reducing the influence of local air from the surface, while less sea ice inhibits this early ascent and allows more mixing of surface air. Variations in sea ice could thus influence the isotopic composition of air masses by controlling the influence of local moisture, which might be recorded only by certain WAIS cores depending on the elevation of the air mass transport and characteristics of the core positions, such as their surrounding topography, elevation, or distance to the coast. Decadal trends or slower changes in these variations could then destroy the recording of the large-scale coherent temperature field by the firn cores and thus cause a loss in spectral signal power in the isotope record towards longer timescales and an increase in the noise level (Fig. 2c, d). In contrast, the East Antarctic Plateau including the DML firn-core sites is higher in elevation and might be more shielded from marine influences by the steep topographic slopes leading to a more coherent signal. This might also explain, besides differences in core quality, the rather low agreement among deep West Antarctic cores on millennial timescales compared to East Antarctic cores (WAIS Divide Project Members, 2013). However, since our inferences are speculative, further studies are needed to help disentangle the role of variations in West Antarctic temperature, atmospheric circulation and sea ice on the recorded isotope variability.

## 5 Conclusions

We presented a simple spectral method to separate the variations recorded by isotope records into a local ("noise") and a common (i.e. spatially coherent) "signal" component. We applied this method to firn cores from the East Antarctic Dronning Maud Land (DML) and the West Antarctic Ice Sheet (WAIS) to estimate, for the first time, the isotopic signal-to-noise ratio (SNR) as a function of timescale. This is of fundamental interest for interpreting isotope records obtained from individual ice cores, since it provides an upper limit on the SNR of the temperature signal recorded by the cores. For DML, the SNR at the interannual timescale is very low, but it steadily increases with timescale reaching values above $0.5$ for multidecadal and slower variations. Therefore, only on these timescales isotope changes from individual cores should be interpreted in terms of regional climate variations. For WAIS, the results are counterintuitive. On interannual to decadal timescales, the estimated SNR is higher than $0.5$, which would support the regional climate interpretation of the isotope records. For longer timescales, however, the estimated SNR decreases, suggesting that local variations start to dominate the recorded isotope variability.

Our method further allows the estimation of the power spectra of the coherent isotope signal. For DML, the spectra of single cores largely resemble white noise. In contrast, the derived signal spectrum shows increasing variability towards longer timescales. Such an increase is also observed in instrumental temperature records and other climate proxies. The marked difference between the raw interpretation of single cores – as it is usually done – and the signal spectra derived from the core

array demonstrates the relevance of the signal and noise separation. The interpretation of the WAIS isotope signal is more challenging, since the signal shows a close-to-constant spectral power even after applying our method. We speculate that this might be due to atmospheric circulation variations that create a local imprint at the different firn-core sites. This might prevent a coherent recording of the large-scale atmospheric temperature field. To test this hypothesis, we suggest to analyse firn-core arrays as a function of the average separation distance between the individual core sites within each array. This could allow us to investigate whether the stable-isotope data record a stronger coherent signal on a regional scale (e.g., $\sim 1$–$10$ km) than on the larger scales analysed here. A similar approach in DML would also help to better constrain the spatial correlation scales of the precipitation intermittency.

We conclude that the pronounced differences seen between East and West Antarctica could thus be related to the differences in the topographic settings and the different marine influence (Noone and Simmonds, 2004; WAIS Divide Project Members, 2013). Attempts to reconstruct the temperature variability in these regions based on firn and ice cores should thus not only aim at averaging as many records as possible, but also consider the spatial coherence of the circulation and precipitation patterns in order to establish an optimal strategy for averaging, or obtaining new, firn-core isotope records. Additionally, extending our analyses to data derived from non-isotope-based temperature proxies could give further insights into the true spectral shape of temperature variability in Antarctica.

Finally, our approach of separating signal and noise from a set of spatially distributed records is also applicable beyond Antarctic ice cores. The challenge of low and timescale-dependent SNR is common to many high-resolution climate archives, and the number of nearby core sites is continuously increasing. Therefore, we envision our approach to better constrain the reliability of proxy records as a function of timescale in general, and to allow a significant improvement of our knowledge of past climate variability.

*Code and data availability.* Software for the spectral analyses, the implementation of the method and the plotting of the results is available as the R package *proxysnr*; source code for the package is hosted in the public git repository at https://github.com/EarthSystemDiagnostics/proxysnr, a snapshot of the code used for this publication (version 0.1.0) is archived under https://doi.org/10.5281/zenodo.2027639 (Münch, 2018a). Software to model the time uncertainty of the isotope records is based on the MATLAB code by Comboul et al. (2014), which has been adapted for this publication and implemented in R as the package *simproxyage*; its source code is available from the public git repository at https://github.com/EarthSystemDiagnostics/simproxyage, a snapshot of the code used for this publication (version 0.1.1) is archived under https://doi.org/10.5281/zenodo.2025833 (Münch, 2018b). The diffusion length calculations have been performed with the R package *FirnR* which is available on request from the authors.

The original DML firn-core and trench oxygen isotope data are archived at the PANGAEA database (https://www.pangaea.de) under https://doi.org/10.1594/PANGAEA.728240 (Graf et al., 2002b) and https://doi.org/10.1594/PANGAEA.876639 (Münch et al., 2017b), respectively. PANGAEA is hosted by the Alfred Wegener Institute Helmholtz Centre for Polar and Marine Research (AWI), Bremerhaven and the Center for Marine Environmental Sciences (MARUM), Bremen, Germany. The original WAIS firn-core oxygen isotope data are archived at the U.S. Antarctic Program Data Center (USAP-DC; http://www.usap-dc.org/) under https://doi.org/10.7265/N5QJ7F8B (Steig, 2013). USAP-DC is hosted by Lamont-Doherty Earth Observatory of Columbia University, Palisades, USA. The processed isotope data as used in

this study, together with generating code, are provided with the package *proxysnr*. The ERA-Interim reanalysis data is available from the European Centre for Medium-Range Weather Forecasts (ECMWF) under http://apps.ecmwf.int/datasets/data/interim-full-moda/levtype=sfc/ (European Centre for Medium-Range Weather Forecasts, 2018). All relevant site parameter data for the diffusion length calculations are available from the publications referenced in Table 1 and, together with the simulated diffusion length estimates, provided with the package
*proxysnr*.

## Appendix A:  Theoretical spectra for site-specific diffusion

We derive the effect of site-specific diffusion on the estimates of the mean spectrum and the spectrum of the stack given a core array consisting of $n$ isotope records.

The spectrum of a time series $x(t)$ is given by the absolute square of its Fourier transformation, $\mathcal{X} = \mathcal{F}(x)\mathcal{F}^*(x)$, where
$\mathcal{F}^*(x)$ denotes the complex conjugate of $\mathcal{F}(x)$. Given an isotope record at site $i$, $x_i(t) = c(t) + \varepsilon_i(t)$, where $c(t)$ denotes the common signal and $\varepsilon_i(t)$ is noise, the time series after diffusion is $x'_i(t) = g_i \star (c(t) + \varepsilon_i(t))$. Here, $\star$ denotes convolution and $g_i$ is a Gaussian convolution kernel whose standard deviation is the diffusion length $\sigma_i$ that is a function of depth (or, equivalently, time since deposition) and depends on site $i$ due to its dependence on local temperature, atmospheric pressure and accumulation rate (Appendix B). The spectrum of $x'$ is then

$$\mathcal{X}_i = \mathcal{F}(g_i)\mathcal{F}(c + \varepsilon_i) \cdot \mathcal{F}^*(g_i)\mathcal{F}^*(c + \varepsilon_i)$$
$$= \mathcal{G}_i \left( \mathcal{C} + \mathcal{N}_i \right), \tag{A1}$$

with $\mathcal{C} = \mathcal{F}(c)\mathcal{F}^*(c)$, $\mathcal{N} = \mathcal{F}(\varepsilon_i)\mathcal{F}^*(\varepsilon_i)$ and where the linear site-specific diffusion transfer function was defined as $\mathcal{G}_i := \mathcal{F}(g_i)\mathcal{F}^*(g_i)$[1]. Assuming that the statistical properties of the individual noise terms are the same for all sites, $\mathcal{N}_i \equiv \mathcal{N}$, the mean across $n$ such spectra is

$$\mathcal{M} = \frac{1}{n} \sum_{i=1}^{n} \mathcal{X}_i = \overline{\mathcal{G}} \left( \mathcal{C} + \mathcal{N} \right) \tag{A2}$$

with the average diffusion transfer function $\overline{\mathcal{G}} := \frac{1}{n} \sum_{i=1}^{n} \mathcal{G}_i$.

The average in the time domain of $n$ independent noise "signals" subject to site-specific diffusion, $y_{\text{noise}}(t) = \frac{1}{n} \sum_{i=1}^{n} g_i \star \varepsilon_i(t)$, has a spectrum of

$$\mathcal{S}_{\text{noise}} = \mathcal{F}(y_{\text{noise}})\mathcal{F}^*(y_{\text{noise}})$$
$$= \frac{1}{n^2} \sum_i \sum_j \mathcal{F}(g_i)\mathcal{F}^*(g_j)\mathcal{F}(\varepsilon_i)\mathcal{F}^*(\varepsilon_j). \tag{A3}$$

We have $\mathcal{F}(\varepsilon_i)\mathcal{F}^*(\varepsilon_j) \neq 0$ only for $i = j$, and hence

$$\mathcal{S}_{\text{noise}} = \frac{1}{n} \overline{\mathcal{G}} \mathcal{N}, \tag{A4}$$

---

[1] We note that there is no closed form expression for $\mathcal{G}_i$; however, in case of a constant diffusion length the transfer function is a Gaussian (van der Wel et al., 2015).

which shows that averaging $n$ independent noise records reduces the spectral power by $1/n$, as expected.

By contrast, the spectrum of the average of $n$ coherent signals subject to site-specific diffusion, $y_{\text{coh}}(t) = \frac{1}{n}\sum_{i=1}^{n} g_i \star c(t)$, is

$$
\begin{aligned}
\mathcal{S}_{\text{coh}} &= \mathcal{F}(y_{\text{coh}})\mathcal{F}^*(y_{\text{coh}}) \\
&= \frac{1}{n^2}\mathcal{F}(c)\mathcal{F}^*(c)\sum_i\sum_j \mathcal{F}(g_i)\mathcal{F}^*(g_j) \\
&= \frac{1}{n}\mathcal{C}\left(\overline{\mathcal{G}} + \frac{1}{n}\sum_{i\neq j}\mathcal{F}(g_i)\mathcal{F}^*(g_j)\right).
\end{aligned}
\tag{A5}
$$

For small differences in the diffusion lengths between the sites, the transfer functions are approximately the same and we can simplify the second term in brackets in Eq. (A5),

$$
\frac{1}{n}\sum_{i\neq j}\mathcal{F}(g_i)\mathcal{F}^*(g_j) \approx \frac{1}{n}\sum_{i\neq j}\mathcal{F}(g_i)\mathcal{F}^*(g_i).
\tag{A6}
$$

From rearranging the summation terms we find that this is identical to $(n-1)\overline{\mathcal{G}}$, and Eq. (A5) becomes

$$
\mathcal{S}_{\text{coh}} \approx \overline{\mathcal{G}}\mathcal{C}.
\tag{A7}
$$

In this approximation, the average over diffused coherent signals does not reduce the spectral power, as one would expect. We tested the approximation for our core arrays by comparing simulation results with surrogate data between the cases of independent and coherent noise. We find that (A7) is a reasonable approximation for the full expression (A5). However, we note that (A5) has slightly less power on the high-frequency end as compared to (A7) since the site-specific diffusion destroys part of the coherence between the sites.

## Appendix B: Estimates of the transfer functions for diffusion and time uncertainty

To estimate the transfer functions for diffusion ($\overline{\mathcal{G}}$) and time uncertainty ($\Phi$), we simulate for each core array $n$ surrogate records of the same time duration as the original isotope records. For each record, we simulate the effects of diffusion and time uncertainty, respectively, and then calculate in each case and for each core array the spectrum of the average record ($\mathcal{S}_{\text{diffusion}}$ or $\mathcal{S}_{\text{time}}$). At the same time, the spectrum of the average record of the initial surrogate data ($\mathcal{S}_0$) is calculated without applying diffusion or time uncertainty. The entire process is repeated $k$ times and the resulting spectra are averaged to reduce the spectral uncertainty. We report $\overline{\mathcal{G}} = \langle\mathcal{S}_{\text{diffusion}}\rangle/\langle\mathcal{S}_0\rangle$ and $\Phi = \langle\mathcal{S}_{\text{time}}\rangle/\langle\mathcal{S}_0\rangle$ as the transfer functions, where $\langle\cdot\rangle$ denotes the average over the $k$ simulations where we use $k = 10^5$.

For the diffusion simulations, each surrogate record is smoothed by convolution with a Gaussian kernel of width $\sigma$, where $\sigma$ is the diffusion length, measured in yr, sensitive to ambient temperature, pressure and the density of the firn (Whillans and Grootes, 1985) and therefore site-specific and a function of depth. We treat the dependency on density according to Gkinis et al. (2014) with diffusivity after Johnsen et al. (2000); firn density is modelled according to the Herron–Langway model (Herron

and Langway, 1980). Surface air pressure is calculated from the barometric height formula using the local elevation and firn temperature. For the surface firn density we assume a constant value of $340 \, \mathrm{kg \, m^{-3}}$ for all sites. Range of input parameters and their references are given in Table 1. Missing temperature information for the cores FB9807, FB9811 and FB9815 are filled with the temperatures from the nearby cores B32 (1 km), FB9812 (19 km) and FB9805 (28 km), respectively. We simulate

diffusion lengths measured in cm for core lengths of 500 m at a resolution of 1 cm, convert the values into yr, and interpolate them onto a regular time axis at a resolution of 0.5 yr. This higher temporal resolution compared to the annual target resolution of the isotope records is necessary for numerical stability of the diffusion results at the high-frequency end. The diffusion transfer functions are then interpolated in frequency space onto the frequency axis of the isotope records. Simulated diffusion lengths at a depth corresponding to 200 yr after deposition of the snow range, across the firn-core sites, from 0.6 to 1.3 yr

(8.3–11.4 cm) for DML and from 0.3 to 0.6 yr (8.9–11.4 cm) for WAIS; for DML and 1000 yr after deposition of the snow diffusion lengths range from 1.1 to 1.6 yr (8.2–9.1 cm).

To simulate the effect of time uncertainty, we apply the modelling approach by Comboul et al. (2014). This model represents an unconstrained process, meaning that the time uncertainty monotonically increases with modelled age, thus deeper in the core. To account for volcanic tie points where the chronologies of the isotope records are fixed (neglecting time uncertainties of

the volcanic chronologies themselves), we modify the Comboul approach by forcing the model back to zero time uncertainty at the reported volcanic tie points (constrained process) in form of a Brownian Bridge process. In contrast to diffusional smoothing, it is not a priori clear whether time uncertainty acts as a linear transfer function on the spectrum of the average record. We tested this by using different spectral characteristics of the surrogate data, adopting the cases of white noise as well as power laws with spectral slopes of $\beta = 0.5, 1, 1.5$ and 2, and found that the modelled effect of time uncertainty is insensitive

to the spectral shape of the input data and indeed acts as a linear transfer function. The Comboul model includes as parameters the rates of missing and double-counted annual layers. We assume equal rates for these processes which we set such so that the yielded maximum time uncertainties (maximum standard deviation over simulated chronologies) match the reported time uncertainties of the isotope records. The reported value of the DML records ($\sim 2 \%$ of the time interval to the nearest tie point, Graf et al., 2002a) erroneously implies that the uncertainty increases linearly with time (Comboul et al., 2014). Here, as a best

guess, we choose a process rate that yields a maximum standard deviation of 2 % for a constrained process of duration equal to the mean interval length between the tie points. In summary, we use process rates of $\gamma = 0.013$ for the DML and of $\gamma = 0.027$ for the WAIS cores, respectively.

The resulting transfer functions as used for the main results are shown in Fig. (B1). Diffusion shows a stronger smoothing for the longer records of DML2 than for DML1 as it had more time to act. WAIS shows less smoothing as compared to DML1

due to the higher accumulation rates. The influence of diffusion is negligible for frequencies below the decadal period for all core arrays. The effect of time uncertainty is negligible for frequencies below the decadal period for DML1 and WAIS but only for frequencies below the 20 yr period for DML2 due to the longer distances between the volcanic tie points.

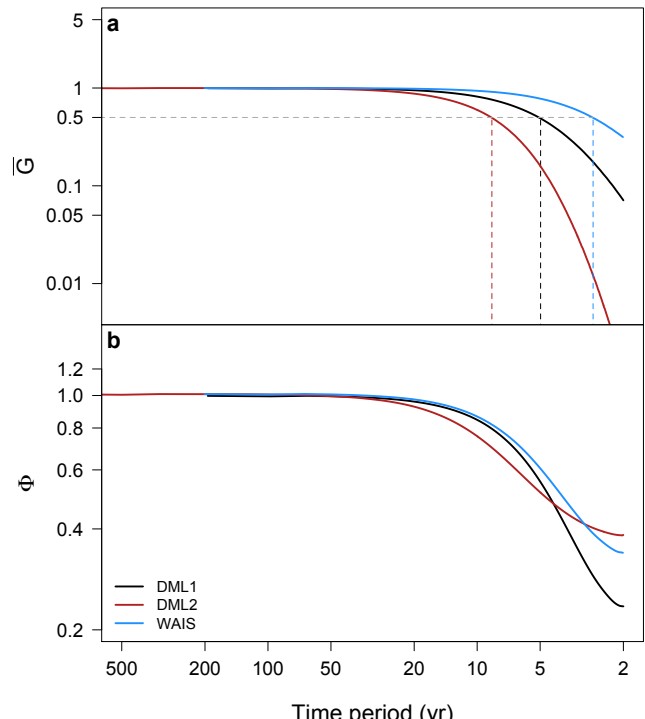

**Figure B1.** Estimates of the spectral transfer functions for the effects of site-specific diffusion (**a**) and time uncertainty (**b**) for the three studied core arrays DML1 (black), DML2 (red) and WAIS (blue). The estimates are based on simulations with surrogate data as explained in the text. Both effects are largely negligible beyond the decadal period. The horizontal dashed line in (**a**) at $0.5$ denotes the chosen transfer function value for constraining the diffusion correction (see main text), corresponding to minimum time periods until which we analyse the respective spectral data as indicated by the vertical dashed lines.

## Appendix C: ERA-Interim temperature and precipitation field decorrelation scales

To estimate the present-day spatial decorrelation scales of the atmospheric temperature and the precipitation fields in our study regions, we resort to ERA-Interim reanalysis data (Dee et al., 2011; European Centre for Medium-Range Weather Forecasts, 2018), since direct observations are sparse. ERA-Interim is in general regarded the most reliable reanalysis product both
5   for temperature and precipitation in Antarctica; the reanalysis temperatures are, despite significant biases in mean values, well correlated with station observations at monthly to interannual timescales (Bracegirdle and Marshall, 2012; Jones and Lister, 2015), and the reanalysis produces the closest match to Antarctic snowfall regarding both mean values and seasonal to interannual variability (Bromwich et al., 2011; Palerme et al., 2017).

To obtain deccorelation scales, we calculate the correlations between reanalysis time series from the gridbox closest to the
10   location of the EDML site ($75°$ S, $0°$ W) with the respective time series of all other gridboxes of the data set south of $60°$ S. Using an exponential fit to the correlations of the form $\exp(-d/\Delta)$, where $d$ is the distance between EDML and a particular

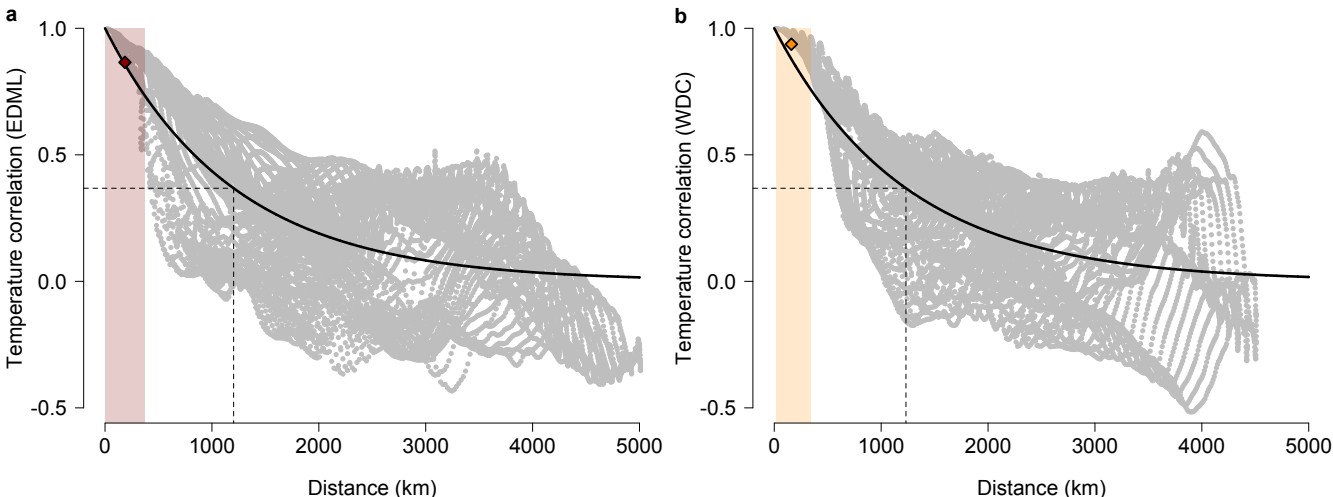

**Figure C1.** Present-day temperature decorrelation across DML and WAIS. Shown are the correlations (grey dots) of ERA-Interim annual-mean temperatures (Dee et al., 2011) at (**a**) EDML and (**b**) WDC with all other gridboxes below $60°$ S. Black lines show an exponential fit to the data; dashed lines indicate the point at which the correlations in the model have dropped to $1/e$ (at $1202\,(1232)$ km for EDML (WDC)). Coloured shading indicates the range of site distances of the studied firn-core arrays (Table 1); the average correlation across each range is marked by the filled diamond (average correlation of $0.87\,(0.94)$ for EDML (WDC)).

gridbox, yields the decorrelation scale through the fit parameter $\Delta$. The same approach is applied to the WDC site ($79.5°$ S, $112°$ W).

For annual mean temperatures, the analysis yields decorrelation scales of around $1200$ km for both regions (Fig. C1). The decorrelation scales of precipation intermittency are expected to depend on the decorrelation scales of the seasonal precipitation distribution rather than on the total annual precipitation amount (Persson et al., 2011). However, our analysis suggests that the precipitation decorrelation scales are actually more ore less insensitive on the chosen variable with estimates between roughly $300$ and $500$ km for both regions for a variety of chosen fields (total precipitation amount, difference between summer (months DJF) and winter (months JJA) precipitation amount, fraction of summer over winter (winter over summer) precipitation amount, and fraction of summer (winter) over total precipitation amount).

*Author contributions.* TM and TL designed the research, developed the methodology and interpreted the results. TM reviewed relevant literature, established the database and performed all analyses. TM wrote a first version of the manuscript which was revised by both authors.

*Competing interests.* The authors declare that they have no conflict of interest.

*Acknowledgements.* We thank all scientists, technicians and the logistic personnel who contributed to the sampling of the firn cores and the measurement of the used stable-isotope data, and we are grateful for making the data publicly available. We are thankful for valuable discussions with and comments by Torben Kunz, Jürgen Kurths, Johannes Freitag and Maria Hörhold. All plots and numerical analyses were carried out using the open-source software R: A Language and Environment for Statistical Computing. This project was supported by Helmholtz funding through the Polar Regions and Coasts in the Changing Earth System (PACES) programme of the Alfred Wegener Institute, by the Initiative and Networking Fund of the Helmholtz Association Grant VG-NH900 and by the European Research Council (ERC) under the European Union's Horizon 2020 research and innovation programme (grant agreement no. 716092). It further contributes to the German BMBF project PalMod. We thank Lukas Jonkers for his kind handling of the manuscript as well as Dmitry Divine and one anonymous referee for their detailed review and helpful comments.

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
