# Peer review of "What climate signal is contained in decadal to centennial scale isotope variations from Antarctic ice cores?"

_Climate of the Past, 2018_

## Referee Comment (RC1) · D. Divine (Referee) · 24 Sep 2018

**Review of a manuscript for** *Climate of the Past*

**What climate signal is contained in decadal to centennial scale isotope variations from Antarctic ice cores?** by Munch and Laepple

**Overall:**

In this manuscript the authors present a method for calculating a timescale dependent SNR for an array of climate proxy records with a common climatic signal and a physical mechanism(s) behind.

For a particular case presented, the study uses ice cores based isotopic records and accounts for the effects of stratigraphic noise, diffusion in firn and timescale uncertainties elaborating the respective power spectral densities for the background climate signal and the aforementioned contributing noise factors. The proposed technique is then applied to ice core arrays from DML and WAIS. Opposite timescale behavior of SNR for the two core networks is linked to the homogeneity/heterogeneity of distillation trajectories between the two regions associated, for example, with the effects of sea ice on isotopes in precipitation.

In general the paper is clearly written and results are well presented. Moreover, my general impression over this study, is that this was one of the rare cases I had so far as a reviewer that can be published almost "as is". When reading the manuscript, I left a number of remarks/suggestions/question marks that I planned to list later when writing this review, yet it turned out in the end that almost all of them the authors have already addressed in Discussion and Conclusions.

This study is certainly recommended for those who deals with multiproxy archives – this is an explicit demonstration of a value of a single proxy (ice core) record and a clear warning against overinterpreting single spikes/events on the shorter timescales. Therefore, I consider the manuscript deserves to be published after some very minor modifications to the content if the authors/editor finds them relevant.

**Minor comments**

Page 2, line 5: "…to a first approximation, changes in isotopic composition are only recorded in the ice if there is snowfall." Recent studies suggest the effects of air (and hence water vapor) exchange across the firn –air interface in between the precipitation events may have a larger impact on the final d18O in snow than previously thought, see for example Stenni et al., 2016 , 10.5194/tc-10-2415-2016. It actually increases the role of SAT variability throughout the accumulation season even given the intermittency of precipitation itself.

Page 6, line 4: Please provide a ref to eq. (7)

Page 6, line 9: "… for display purposes… smoothed using a Gaussian kernel". Still the motivation is not clear, would it be possible to see an unsmoothed signal (in the letter of response for example). What is the kernel bandwidth used?

Page 11, lines 10-20. Quality of ERA precipitation needs to be briefly discussed. How reliable are the estimates based on this variable?

Page 14, lines 30-33. The effects of sea ice on the modelled isotopic composition of precipitation in Antarctica can be found in the studies by Noone , e.g. Noone, D., and I. Simmonds (2004), Sea ice control of water isotope transport to Antarctica and implications for ice core interpretation, J. Geophys. Res., 109, D07105, doi:10.1029/2003JD004228. The authors are recommended to see if these results can be used to elaborate more on the potential controls of the different patterns in SNR found between the two study regions.

Page 19, lines 3-5. The authors present the winter and summer precipitation results. It is highly recommended to do the same analysis for the fall and spring seasons. The semi-annual oscillation (SAO) tends to modulate the seasonal distribution of precipitation depending on the strength of the semiannual harmonic. In addition, for West Antarctica (though shown for Faraday only in Broeke et al., 2000, part 4) the sea ice extent in the Amundsen and Bellingshausen seas (also linked with SAO strength) was shown to modulate the seasonal precipitation too. One can speculate that a long term variability in the strength/position of the low in contraction phase of the SAO (March and September) can actually be one of the mechanisms responsible for disruption of the coherence between the isotopic records on the longer timescales.

See in the series of earlier publications by Van den Broeke

Van den Broeke, M.R. 1998a. 'The semiannual oscillation and Antarctic climate, part 1: influence on near-surface temperatures (1957–1979)', *Antarctic Sci*., **10**(2), 175–183.
Van den Broeke, M.R. 1998b. 'The semiannual oscillation and Antarctic climate, part 2: recent changes', *Antarctic Sci*., **10**(2), 184–191.
Van den Broeke, M.R. 2000. 'The semiannual oscillation and Antarctic climate, part 3: the role of near-surface wind speed and cloudiness', *Int. J. Climatol*., **20**(2), 117–13
Van Den Broeke, M. (2000), The semi-annual oscillation and Antarctic climate. Part 4: a note on sea ice cover in the Amundsen and Bellingshausen Seas. Int. J. Climatol., 20: 455-462. doi:10.1002/(SICI)1097-0088(20000330)20:4<455::AID-JOC482>3.0.CO;2-M

---

## Referee Comment (RC2) · Anonymous Referee #2 · 28 Sep 2018

\*

[Figure]

**Review of the article: 'What climate signal is contained in decadal to centennial scale isotope variations from Antarctic ice cores?' by Thomas Münch and Thomas Laepple, submitted to Climate of the Past.**

September 28, 2018

**1   General comments**

The article submitted deals with quantification of climate signal versus noise in ice cores from Antarctica. It is therefore well within the scope of Climate of the Past, and addresses an important issue for climatologists. Its aim is not to present new ice core data, but to present a methodology to evaluate (quantify) the climate signal contained in a series of records.

The methodology is based on a spectral analysis of the dataset, where the spectrum of the stacked record is compared to the mean spectrum and to white noise. The method also includes a correction for diffusion and for time uncertainty. The Methodology section is concise, because details are described in Appendixes. The paragraph 3.1 in

Results is a useful complement to the methodology section, as it applies the method to an example, and provides a figure where the various steps are represented. It is well suited to an article that aims a large audience, not necessarily with statistical background, and who might overlook the equations in the methodology section and Appendixes.

In the continuation of the Results section, the figures are described in less details. Some more precision is needed here, so that the main message is not obscured by unanswered questions on the parts of the figures that are not described. The results are different for the two studied regions. At EDML, the signal to noise ratio is found to increase for longer time scales (0.2 to 1), whereas at WAIS, it is relatively stable, and even seems to decrease at long (centennial) time scales. For the first region, the authors therefore recommend to use single cores only for multidecadal or longer timescales. For the second region, oppositely, they conclude that single cores yield good regional information at interannual and decadal scales, but give a more local information at longer time scales.

In the Discussion, the authors consider the possible contribution of four processes to the climate signal, by looking at their spatial scale of coherence. They note that precipitation intermittency acts as noise or contributes to signal, depending on the scale considered.
They also discuss the unexpected decrease of signal power at WAIS for longer timescales. This coastal region is particularly sensitive to the variability in atmospheric circulation. They suggest that slow processes modifying the topography of the region may reduce the spatial coherence of the signal over long timescales.

The conclusions of the article are important regarding the confidence that we can attribute to one or several ice core records. The results for WAIS are unexpected, and

therefore may trigger more research in the area, or allow to consider differently the results from previous drillings. Lastly, as noted by the authors, the new methodology can be applied to other records, and therefore be of interest to a large audience.

The manuscript is well written and well structured, with a good use of appendixes to procure relevant information. Given that the conclusions of the article are interesting, and well supported by the Method and Results sections, I recommend that this article should be accepted by Climate of the Past with minor corrections.

**2 Specific comments**

P2, l12: 'the diffusion of vapor through the firn column smoothes the isotope variations, reducing the power spectral density at high frequencies'
The first part of the sentence is very clear but the second part is less straightforward. Is it possible to add an intermediary step? '…smoothes the isotope variations, with stronger impact at short distances and therefore on the high frequencies.'

P2, l. 24: 'Furthermore, knowing the timescale dependence, and thus the spectral shape of the noise, would allow the correction of the isotope-inferred variability estimates for the noise contribution.'
This sentence is complicated, it takes some time to find the subject. Is it possible to split it or reverse it?
'Furthermore, in order to correct the noise contribution to the isotope inferred variability estimates, it is necessary to know the noise distribution over various timescales, i.e. its spectral shape.'

P3, l14: 'The time uncertainty of the record chronologies, . . ., has been reported with 2 % of the time interval to the nearest tie point.'
This sentence is unclear, is there something missing? Is it 'has been reported to be 2 % of the time interval'?

P5, l8: Equations 4
Although it was only a few lines above, please repeat here what M and S are (respectively the mean of spectra, and the spectra of the stacked record along time). Also, is it possible to get another symbol for the noise (or at least a very specific font) so that it is easier to distinguish between N and N?

P5, l. 19: 'we restrict our analyses to the frequency region where G<2. This avoids large uncertainties and yields cutoff frequencies of. . .'
Based on Figure B1 in annex, G is always below 2. Its maximum value is one. This seems logical since a filter will eliminate some frequencies, not add more frequencies to the signal. Should it be 'restrict to the region where G>0.5'? (so that the short frequencies will not triple in power after correction)?
Please clarify this point in the text or in the Appendix B.
Please define the term 'cut-off frequency' (and maybe draw the limits on Figure B1).

P6, l2: 'the value of F is related to the correlation between. . .'
Is the correlation r providing more information than F (on the quality of the data) or the same information? Is the r value only for intercomparing with other studies?

P6, l20-28:

1. Is it possible to insert here the symbols used previously in the methodology section (M, S) to facilitate the identification of the various terms?

2. The mean over all individual spectra, M, (figure1, black line) . . .

3. The mean spectrum divided by the number of records (M/15, figure 1, dashed line) . . .

4. . . .averaging across record that contain noise and additionally. . . (S. . ., Figure 1, brown line)

Furthermore, please precise 'averaging across records along the time dimension' or 'in time space' to refer to the 'time stack' S, by opposition to the spectrum mean M.

P6, l29: 'For short timescales (2 to seven years), . . ., is consistent with the null hypothesis of independent noise.'
The curves are closest between 3 and 5 years; they diverge again around 2. What causes this divergence from white noise? Is it an artefact?

P7, l6-7: 'Unlike the average spectrum across all individual isotope variations (fig1), the corrected DML1 signal spectrum shows an increase of power spectral density with increasing time scale (fig. 2a)'.

1. Please precise which line is described here (color, symbol).

2. Both the black and brown line show an increase of power spectral density towards higher timescales on Fig1 (strongest between 2 and 20, then flat area at longer timescales). This proposition has to be nuanced.

3. For the brown line, there is again an increase at very long timescales (>50) still on Figure 1.

4. On Figure 2, the 3 DML1 curves are very similar for t>20. So, the correction does not seem to have a huge effect. Moreover, the increase of power between 50 and 200 years exist with/ without correction...

5. Lastly, regarding the increase between 3 and 20 years on Figure 2, the correction actually reduces the slope...

Possible correction to this sentence:
'Unlike the average spectrum across all individual variations (M, black line on Fig 1), the corrected DML1 signal spectrum (C, solid blue line on Figure 2a) shows an increase of power spectral densities with increasing time scales at time scales larger than 50 years.'

P7, l. 7-8:'This is confirmed by the three 1000-yr long records from the DML2 data set whose signal exhibits a similar power spectrum in the range of timescales that overlap.' The two curves are not really a perfect match.
There is common behavior between 60 and 100 years (increasing) and between 5 and 15 years (increasing) but the middle part (15 to 60) is different between the two curves. DML1 is slowly decreasing (or flat), while DML2 is strongly decreasing until 30, then strongly increasing. Maybe add: '...for time scales longer than 50 years.'
Or split the paragraph in two, one dealing with long time scales, and the other with short time scales.

P9, l.6: '...decrease in signal power on centennial timescales'
Is it possible that this decrease is an artefact? How certain are we of the power of the 100-year frequency on a 200-year record?

P11, l1: 'In fact, the average correlation of 0.87 (0.94) over distances up to the maximal

intercore distance...'
Please add here a reference to Appendix C, Figure C1. Otherwise the origin of these numbers (and their meaning) is unknown.
Based on the figure caption, this correlation is an average (for various intercore distances). Thus, maybe the formulation 'up to the maximum intercore distance' is misleading here. Why not say the correlation for average distance between cores instead?

P12, l. 11-13; 'The raw noise spectra derived from the two DML data sets exhibit a clear imprint from the diffusional smoothing in the firn, as suggested by the smaller PSD for periods <20 years of the raw DML2 spectrum, i.e. prior to correction, in comparison to DML1 (fig. 2b).'
This is too fast. Please split in two steps:

1. first, raw to corrected for both records (DML1 and DML2): the effect of diffusion is to decrease power at high frequencies;

2. second, comparison of the raw data for DML1 and DML2: DML2 is more affected by diffusion, since frequencies between 12 and 30 are affected (low power) only for DML2.

P12, l30: 'yields a slope of the DML2 signal spectrum of roughly beta=0.6'
Is it possible to add this fit to Figure 2 or 3?

P13, l. 7: 'minimum averaging period constrained by the diffusion correction ( 2.5 years)'
Is it correct to compare r values at different averaging times (1 year and 2.5 years)? Could this also contribute to the higher correlation compared to previous study of the

same data?

P14, l. 1-3: 'For WAIS, the higher SNR at interannual timescales as compared to DML is consistent with the general notion that higher accumulation rates result in higher SNR.'

1. Please insert the SNR values for both.

2. For N=1 and deltat=5 years, r=0.45 at DML and r=0.51 at WAIS. The difference looks very small (especially considering the 3-times accumulation at WAIS). Why is the increase in SNR not scaled to the large increase in accumulation?

P14, l26-28: 'Together with the observed spectral shapes on the longer timescales (30-100y) our results therefore might indicate a true increase in local variability at the WAIS sites towards longer time scales, and a close-to-constant or even decreasing coherent signal variability.'
This sentence is strange. It looks unbalanced with a first part dealing with long time scales and a second part dealing with... probably shorter time scales?
Is it possible to correct this sentence?
To facilitate reading, maybe this paragraph could be limited to the issue of high frequencies at WAIS, since the discussion for the long timescales already existed in the previous one.
Is it possible to separate the issue of high frequencies for noise and for signal?

What are the hypotheses explaining this decrease in noise at high frequencies? Could it be something like wind blowing that would homogenize only the surface over large areas? Or large-scale evaporation of surface snow homogenizing its composition?

**2.1 Appendix A**

P16, l 23: 'a Gaussian convolution kernel whose standard deviation is the diffusion
length $\sigma_i$ that is a function of depth (time) and depends on site i.'
This is very synthetic, but not very clear. Please add a reference to Appendix B where
more details are provided. Please give some values of $\sigma_i$ (and possibly the associated
frequencies) at various depths. Is there a formula that relates $\sigma_i$ to time (for each site)?
If not, the use of the term 'function' might be misleading here. Indeed, $\sigma_i$ seems to
answer on density, temperature and pressure, and therefore to be more similar to an
adjustable parameter in the densification model.

**2.2 Appendix C**

P19, l8: This sentence is complicated. It would be simpler to describe the procedure
at EDML, and then say that the same method is applied to WAIS.

**3 Technical corrections**

1. P5, l. 12: "inidividual" typo

2. P6, l. 14: Please remove 'exemplarily' which is redundant with 'to illustrate our
   method'.

3. P11, l. 8: 'estimated signal term' Please add a reference to the equation (5 or 6).

4. P16, l. 12: 'core sites' typo

5. P17, l.8: It is $F(\varepsilon_i)F(\varepsilon_j) \neq 0$ only for i=j, and hence... Maybe 'we have' instead of 'it is'?

---

## Editor Comment (EC1) · L. Jonkers (Editor) · 12 Oct 2018

Thank you for your contribution to the inter-journal special issue 'Paleoclimate data synthesis and analysis of associated uncertainty' and for your help to promote open source paleoclimate science.

One of the goals of the special issue on 'Paleoclimate data synthesis and analysis of associated uncertainty' is to promote good data stewardship in paleoclimatology. Therefore, the data handling of all contributions to the special issue will be reviewed independently from the normal peer reviews and short comments. While we realise that this may lead to additional work on the authors' side, we believe that good data

stewardship is essential to guarantee transparency and reproducibility of the results as well as to promote the reuse of the data. The editors will be requesting evidence that all data presented in the submissions are made available freely and adhere to the FAIR concept (https://www.nature.com/articles/sdata201618). This applies to original data, as well as to data compilations and derived data products. Where relevant, authors are asked to adhere to the practice of attributing data to original authors through data citation and encouraged to share code used to treat original data and generate derived data products.

Specific comments for 'What climate signal is contained in decadal to centennial scale isotope variations from Antarctic ice cores?' By Muench and Laepple.

Even though the manuscript is based on data that have already been published elsewhere, the following points will add to the reproducibility of your results:

- Please include a data availability statement, including data citations, for the data used (DML and WAIS ice cores and ERA Interim reanalysis data).

- Please consider to make code used in your study publicly available in online repository.

With kind regards,

Lukas Jonkers On behalf of the editorial team.

---

## Author Comment (AC1) · 29 Oct 2018

Author Reply to the Review Comments by Dmitry Divine (Referee #1)

on the manuscript

cp-2018-112: What climate signal is contained in decadal to centennial scale isotope variations from Antarctic ice cores?
by Thomas Münch and Thomas Laepple

Thank you very much, Dmitry Divine, for the time you spent on reading and reviewing our manuscript; we are happy about the positive outcome. Below we include a point-by point response to both the general and to all specific comments. The original referee comments are set in normal font, our answers (author comment, AC) are typeset in *green italic font*.
* * *
**Overall:**

In this manuscript the authors present a method for calculating a timescale dependent SNR for an array of climate proxy records with a common climatic signal and a physical mechanism(s) behind.

For a particular case presented, the study uses ice cores based isotopic records and accounts for the effects of stratigraphic noise, diffusion in firn and timescale uncertainties elaborating the respective power spectral densities for the background climate signal and the aforementioned contributing noise factors. The proposed technique is then applied to ice core arrays from DML and WAIS. Opposite timescale behavior of SNR for the two core networks is linked to the homogeneity/heterogeneity of distillation trajectories between the two regions associated, for example, with the effects of sea ice on isotopes in precipitation.

In general the paper is clearly written and results are well presented. Moreover, my general impression over this study, is that this was one of the rare cases I had so far as a reviewer that can be published almost "as is". When reading the manuscript, I left a number of remarks/suggestions/question marks that I planned to list later when writing this review, yet it turned out in the end that almost all of them the authors have already addressed in Discussion and Conclusions.

This study is certainly recommended for those who deals with multiproxy archives – this is an explicit demonstration of a value of a single proxy (ice core) record and a clear warning against overinterpreting single spikes/events on the shorter timescales. Therefore, I consider the manuscript deserves to be published after some very minor modifications to the content if the authors/editor finds them relevant.

*AC:*
*Thank you very much for this positive evaluation of our work. We are happy that you find our study relevant, well presented and deserving publication.*

**Minor Comments:**

Page 2, line 5: "…to a first approximation, changes in isotopic composition are only recorded in the ice if there is snowfall." Recent studies suggest the effects of air (and hence water vapor) exchange across the firn –air interface in between the precipitation events may have a larger impact on the final d18O in snow than previously thought, see for example Stenni et al., 2016 , 10.5194/tc-10-2415-2016. It actually increases the role of SAT variability throughout the accumulation season even given the intermittency of precipitation itself.

*AC:*
*We agree that there are indications that vapour exchange processes possibly further affect the isotopic composition of the surface snow and, if connected to SAT variations, could actually increase (again) the role of SAT variability for the isotope variability. Nevertheless, we could show for the Kohnen Station region in DML that, even after averaging away the local stratigraphic noise, the near-surface isotope variability is still very discrepant from the local interannual SAT varibility (Münch et al. 2017). This suggests either precipitation intermittency to be a main driver of the remaining (after reducing the*

*stratigraphic noise) isotope variability, or isotope modifications occurring directly at the surface which are not controlled by SAT variations.*

*In order to briefly reflect this discussion in the manuscript, we suggest to change the respective sentence to: "To a first approximation, changes in isotopic composition are only recorded in the ice if there is snowfall, while the role of water vapour exchange processes in between precipitation events is still debated (Steen-Larsen et al., 2011; Stenni et al., 2016; Casado et al., 2018; Ritter et al, 2016; Münch et al, 2017)."*

Page 6, line 4: Please provide a ref to eq. (7).

*AC:*
*The expression directly follows from the definition of the correlation coefficient as shown in Fisher et al. (1985). We will add this reference to the paper here.*

Page 6, line 9: "… for display purposes… smoothed using a Gaussian kernel". Still the motivation is not clear, would it be possible to see an unsmoothed signal (in the letter of response for example). What is the kernel bandwidth used?

*AC:*
*We agree that the formulation in the respective sentence was unclear. We do not apply the smoothing only for display purposes. In general, a strong smoothing is necessary in order to improve the quality of the spectral estimates. If one assumes that the spectrum of the climate signal (which one aims to reconstruct with proxy data) can be described by piecewise power laws, smoothing in logarithmic space is reasonable, which we have adopted here. This logarithmic smoothing is applied by taking weighted averages of spectral power over a Gaussian smoothing window with a scale factor in log units (Kirchner et al., 2005) which we choose to be 0.1 for the WAIS data and 0.15 for the DML data. The scaling in log units is proportional to the frequency at which the smoothing is applied, thus, at the higher frequencies more data points are averaged resulting in a stronger smoothing. We will add a more thorough motivation for the logarithmic smoothing at this point of the paper.*

*Below you see an unsmoothed version of paper figure 1 where you can observe the strong spectral uncertainty when no smoothing is applied.*

[Figure]

**Figure 1:** *As Fig. 1 from the paper but without logarithmic smoothing applied to the spectral estimates.*

Page 11, lines 10-20. Quality of ERA precipitation needs to be briefly discussed. How reliable are the estimates based on this variable?

*AC:*
*We agree with the reviewer that the quality of the used ERA reanalysis data should be discussed but we think that the appropriate place for this would be the beginning of Appendix C. However, since our results do not critically depend on the accuracy of the estimated decorrelation scales, we suggest to only include a short and general discussion of the ERA-Interim quality of temperature and precipitation in Antarctica at the beginning of Appendix C.*

Page 14, lines 30-33. The effects of sea ice on the modelled isotopic composition of precipitation in Antarctica can be found in the studies by Noone , e.g. Noone, D., and I. Simmonds (2004), Sea ice control of water isotope transport to Antarctica and implications for ice core interpretation, J. Geophys. Res., 109, D07105, doi:10.1029/2003JD004228. The authors are recommended to see if these results can be used to elaborate more on the potential controls of the different patterns in SNR found between the two study regions.

*AC:*
*Thank you very much for pointing us at this work. We will include a reference to Noone and Simmonds and a short discussion of their findings at this point of the manuscript in order to improve our discussion of the role of sea ice changes for the isotopic composition of precipitation.*

Page 19, lines 3-5. The authors present the winter and summer precipitation results. It is highly recommended to do the same analysis for the fall and spring seasons. The semi-annual oscillation (SAO) tends to modulate the seasonal distribution of precipitation depending on the strength of the semiannual harmonic. In addition, for West Antarctica (though shown for Faraday only in Broeke et al., 2000, part 4) the sea ice extent in the Amundsen and Bellingshausen seas (also linked with SAO strength) was shown to modulate the seasonal precipitation too. One can speculate that a long term variability in the strength/position of the low in contraction phase of the SAO (March and September) can actually be one of the mechanisms responsible for disruption of the coherence between the isotopic records on the longer timescales.

See in the series of earlier publications by Van den Broeke.

*AC:*
*Thank you very much for pointing us at these interesting results. We extended the decorrelation analysis of the ERA-Interim precipitation data for the fall and spring seasons. This, however, does not change the results: also for the distribution of spring or fall precipitation (e.g., spring minus fall or spring/total precipitation, etc.) the decorrelation scales lie roughly between 300 and 500 km for both regions. Thus, whatever season drives the precipitation intermittency, the typical spatial scale of the intermittency should be of the order of 300-500 km, and thus a factor of 3-4 below that of the temperature fields.*

*Thomas Münch and Thomas Laepple*

References to the author comments:

Casado, M., et al., The Cryosphere, 12, 1745–1766, https://doi.org/10.5194/tc-12-1745-2018, 2018.

Fisher, D. A., et al., Ann. Glaciol., 7, 76–83, https://doi.org/10.1017/S0260305500005942, 1985.

Kirchner, J. W., Phys. Rev. E, 71, 066110, https://doi.org/10.1103/PhysRevE.71.066110, 2005.

Münch, T., et al., The Cryosphere, 11, 2175–2188, https://doi.org/10.5194/tc-11-2175-2017, 2017.

Ritter, F., et al., The Cryosphere, 10, 1647–1663, https://doi.org/ 10.5194/tc-10-1647-2016, 2016.
Steen-Larsen, H. C., et al., J. Geophys. Res., 116, D06108, https://doi.org/10.1029/2010JD014311, 2011.

Stenni, B., et al., The Cryosphere, 10, 2415–2428, https://doi.org/ 10.5194/tc-10-2415-2016, 2016.

---

## Author Comment (AC2) · 29 Oct 2018

Author Reply to the Review Comments by an anonymous reviewer (Referee #2)

on the manuscript

cp-2018-112: What climate signal is contained in decadal to centennial scale isotope variations from Antarctic ice cores?
by Thomas Münch and Thomas Laepple

We appreciate a lot the very thorough and detailed reading and reviewing of our mansucript by the anonymous reviewer. The many comments and suggestions will be of great help to improve our first version of the paper. Below we include a point-by point response to both the general and to all specific as well as technical comments raised by the referee. The original referee comments are set in normal font, our answers (author comment, AC) are typeset in *green italic font*.
* * *
**General comments:**

The article submitted deals with quantification of climate signal versus noise in ice cores from Antarctica. It is therefore well within the scope of Climate of the Past, and addresses an important issue for climatologists. Its aim is not to present new ice core data, but to present a methodology to evaluate (quantify) the climate signal contained in a series of records.

The methodology is based on a spectral analysis of the dataset, where the spectrum of the stacked record is compared to the mean spectrum and to white noise. The method also includes a correction for diffusion and for time uncertainty. The Methodology section is concise, because details are described in Appendixes. The paragraph 3.1 in Results is a useful complement to the methodology section, as it applies the method to an example, and provides a figure where the various steps are represented. It is well suited to an article that aims a large audience, not necessarily with statistical background, and who might overlook the equations in the methodology section and Appendixes.

*AC:*
*We are happy that the structure of the manuscript and the steps at which we present the results are positively evaluated. Since not every reader of CP might be familiar with the details of the applied spectral analyses, it is indeed our intention to still present our results in a way that is understandable to a broader audience, as we envision our approach to be applicable in many fields of paleoclimatology/proxy research.*

In the continuation of the Results section, the figures are described in less details. Some more precision is needed here, so that the main message is not obscured by unanswered questions on the parts of the figures that are not described. The results are different for the two studied regions. At EDML, the signal to noise ratio is found to increase for longer time scales (0.2 to 1), whereas at WAIS, it is relatively stable, and even seems to decrease at long (centennial) time scales. For the first region, the authors therefore recommend to use single cores only for multidecadal or longer timescales. For the second region, oppositely, they conclude that single cores yield good regional information at interannual and decadal scales, but give a more local information at longer time scales.

*AC:*
*Thank you for these comments. Also in the light of what was said above, we will go through the second part of the results section again to ensure that the results are described in adequate precision in order to be comprehensible for a broad audience; see also our answers to the detailed comments.*

In the Discussion, the authors consider the possible contribution of four processes to the climate signal, by looking at their spatial scale of coherence. They note that precipitation intermittency acts as noise or contributes to signal, depending on the

scale considered.
They also discuss the unexpected decrease of signal power at WAIS for longer timescales. This coastal region is particularly sensitive to the variability in atmospheric circulation. They suggest that slow processes modifying the topography of the region may reduce the spatial coherence of the signal over long timescales.

The conclusions of the article are important regarding the confidence that we can attribute to one or several ice core records. The results for WAIS are unexpected, and therefore may trigger more research in the area, or allow to consider differently the results from previous drillings. Lastly, as noted by the authors, the new methodology can be applied to other records, and therefore be of interest to a large audience.

The manuscript is well written and well structured, with a good use of appendixes to procure relevant information. Given that the conclusions of the article are interesting, and well supported by the Method and Results sections, I recommend that this article should be accepted by Climate of the Past with minor corrections.

*AC:*
*We appreciate this positive evaluation.*

**Specific comments:**

P2, l12: 'the diffusion of vapor through the firn column smoothes the isotope variations, reducing the power spectral density at high frequencies'
The first part of the sentence is very clear but the second part is less straightforward. Is it possible to add an intermediary step? '...smoothes the isotope variations, with stronger impact at short distances and therefore on the high frequencies.'

*AC:*
*We will add the additional information, as suggested, and split the sentence into two parts for the sake of clarity: "Finally, once the snow is deposited, the diffusion of vapour through the firn column smoothes the isotope variations. This has a larger impact on short distances in the firn and therefore reduces the power spectral density of the variations strongest at the high frequencies (ref), which substantially shapes the isotope variability (ref)."*

P2, l. 24: 'Furthermore, knowing the timescale dependence, and thus the spectral shape of the noise, would allow the correction of the isotope-inferred variability estimates for the noise contribution.'
This sentence is complicated, it takes some time to find the subject. Is it possible to split it or reverse it?
'Furthermore, in order to correct the noise contribution to the isotope inferred variability estimates, it is necessary to know the noise distribution over various timescales, i.e. its spectral shape.'

*AC:*
*We will change the respective sentence to: "Furthermore, in order to correct the isotope inferred variability estimates for the noise contribution, it is necessary to know the variance of the noise across timescales, i.e., its spectral shape."*

P3, l14: 'The time uncertainty of the record chronologies, ..., has been reported with 2 % of the time interval to the nearest tie point.'
This sentence is unclear, is there something missing? Is it 'has been reported to be 2 % of the time interval'?

*AC:*
*The sentence indeed was misleading and it should read "...reported to be 2% of the time interval". For the sake of clarity, we will correct this and also rearrange the sentence to: "The record chronologies were established from seasonal layer counting of chemical impurity records constrained by tie points from the dating of volcanic ash layers (Graf et al., 2002a); their uncertainty has been reported to be 2 % of the time interval to the nearest tie point (Graf et al., 2002a)."*

P5, l8: Equations 4
Although it was only a few lines above, please repeat here what M and S are (respectively
the mean of spectra, and the spectra of the stacked record along time). Also, is
it possible to get another symbol for the noise (or at least a very specific font) so that it
is easier to distinguish between N and N?

*AC:*

*We will repeat the definition of M and S here, as suggested, and will change the symbol for the number
of records from upper case 'N' to lower case 'n' throughout the manuscript. We also note here that in
Eq. (2), the lower limit of the sum was erroneously given by "i=0" instead of the correct "i=1". This
will be corrected.*

P5, l. 19: 'we restrict our analyses to the frequency region where G<2. This avoids
large uncertainties and yields cutoff frequencies of...'
Based on Figure B1 in annex, G is always below 2. Its maximum value is one. This
seems logical since a filter will eliminate some frequencies, not add more frequencies
to the signal. Should it be 'restrict to the region where G>0.5'? (so that the short
frequencies will not triple in power after correction)?
Please clarify this point in the text or in the Appendix B.
Please define the term 'cut-off frequency' (and maybe draw the limits on Figure B1).

*AC:*

*We apologize for the confusion. At this point of the manuscript we mixed up $\overline{G}$ and $\overline{G}^{-1}$ (the inverse of
$\overline{G}$, which is actually applied as the correction function in Eqs. (4)). We will correct the sentence to
"restrict to the region where $\overline{G} \geq 0.5$" and clarify in the appendix that this means a correction factor
less than 2 in Eqs. (4). Additionally, we will define the phrase "cutoff frequencies" by paraphrasing the
sentence as "This avoids large uncertainties and translates to a maximum frequency that is used for the
analyses (hereafter: cutoff frequency) of... (Fig. B1)", and we will add the respective frequency limits
as vertical lines in Fig. (B1a).*

P6, l2: 'the value of F is related to the correlation between...'
Is the correlation r providing more information than F (on the quality of the data) or the
same information? Is the r value only for intercomparing with other studies?

*AC:*
*The correlation is mathematically equivalent to the value of $\overline{F}$; however, we think the correlation in
addition provides a more direct and intuitive way of expressing the amount of variability contained in a
stacked isotope record that is related to the common (climate) signal in relation to its total variability.
In order to stress the equivalence of both quantities, we will rephrase the sentence to "The value of $\overline{F}$ is
used to obtain the correlation between the time series of the common signal $c$ and a "stacked" record
$\overline{x}$ built from the spatial average of $n$ individual records:".*

P6, l20-28:

1. Is it possible to insert here the symbols used previously in the methodology sec-
tion (M, S) to facilitate the identification of the various terms?

2. The mean over all individual spectra, M, (figure1, black line)...

3. The mean spectrum divided by the number of records (M/15, figure 1, dashed
line)...

4. ...averaging across record that contain noise and additionally... (S..., Figure 1,
brown line)

*AC:*
*We will insert the respective symbols here, as suggested.*

Furthermore, please precise 'averaging across records along the time dimension' or 'in time space' to refer to the 'time stack' S, by opposition to the spectrum mean M.

*AC:*
*We will clarify the sentence as follows: "In comparison, averaging in the time domain across records that contain noise and additionally a common (i.e. spatially coherent) signal, ...".*

P6, l29: 'For short timescales (2 to seven years), ..., is consistent with the null hypothesis of independent noise.'
The curves are closest between 3 and 5 years; they diverge again around 2. What causes this divergence from white noise? Is it an artefact?

*AC:*
*This divergence from the white noise level close to the 2-yr Nyquist period is probably indeed an artefact caused by noise added in the measurement process, i.e. from the measurement uncertainty and more likely from the "jitter error", thus uncertainty in the definition of annual depth increments upon dating which translates into uncertainty of the annual averages. We will add a respective remark to the manuscript here.*

P7, l6-7: 'Unlike the average spectrum across all individual isotope variations (fig1), the corrected DML1 signal spectrum shows an increase of power spectral density with increasing time scale (fig. 2a)'.

1. Please precise which line is described here (color, symbol).

*AC:*
*We will add the respective descriptions.*

2. Both the black and brown line show an increase of power spectral density towards higher timescales on Fig1 (strongest between 2 and 20, then flat area at longer timescales). This proposition has to be nuanced.

*AC:*
*Please note that both these spectra in Fig1 have not been corrected for the loss of spectral power by diffusion, so most of the increase in spectral power between the 2 and 20 year period is attributable to the decreasing diffusional smoothing towards longer timescales. What we wanted to stress at this part of the manuscript is the difference in spectral shape between the mean spectrum (M, black line in Fig1) and the estimated signal (blue for DML1 in Fig2) for longer timescales, which stems from correcting the isotope variability for the noise contribution. We will rewrite the respective sentence to clarify this.*

3. For the brown line, there is again an increase at very long timescales (>50) still on Figure 1.

*AC:*
*This is indeed expected given that the estimated signal (blue line in Fig2), which is contained in the spectrum of the stack (see Eq. 3), increases in power towards the longer timescales.*

4. On Figure 2, the 3 DML1 curves are very similar for t>20. So, the correction does not seem to have a huge effect. Moreover, the increase of power between 50 and 200 years exist with/ without correction...

*AC:*
*It is indeed expected for the diffusion and time uncertainty corrections to have no effect for periods >20 years, given their estimated transfer functions (Fig. B1). What is still relevant to correct on the longer timescales is the residual amount of noise still contained in the spectrum of the stack (since we only average a finite number of records). This residual noise is subtracted when calculating (solving for) the signal spectrum in Eq. (4a).*

5. Lastly, regarding the increase between 3 and 20 years on Figure 2, the correction actually reduces the slope...

*AC:*
*This is true and already noted in the manuscript (p.7 ll.11-13).*

Possible correction to this sentence:
'Unlike the average spectrum across all individual variations (M, black line on Fig 1), the corrected DML1 signal spectrum (C, solid blue line on Figure 2a) shows an increase of power spectral densities with increasing time scales at time scales larger than 50 years.'

*AC:*
*We will add the respective descriptions, as suggested, and rewrite the sentence to clarify that the corrections include both the correction for residual noise as well as diffusion+time uncertainty, and that they lead to a more steady increase in signal power as compared to the mean spectrum. For further clarification, we will in addition change the legend of the dashed line in Fig. 1a from "Raw" to "Uncorrected signal".*

P7, l. 7-8:'This is confirmed by the three 1000-yr long records from the DML2 data set whose signal exhibits a similar power spectrum in the range of timescales that overlap.' The two curves are not really a perfect match.
There is common behavior between 60 and 100 years (increasing) and between 5 and 15 years (increasing) but the middle part (15 to 60) is different between the two curves. DML1 is slowly decreasing (or flat), while DML2 is strongly decreasing until 30, then strongly increasing. Maybe add: '...for time scales longer than 50 years.'
Or split the paragraph in two, one dealing with long time scales, and the other with short time scales.

*AC:*
*We acknowledge that the two curves are indeed not a perfect match, but only show a similar slope. We will weaken the statement accordingly.*

P9, l.6: '...decrease in signal power on centennial timescales'
Is it possible that this decrease is an artefact? How certain are we of the power of the 100-year frequency on a 200-year record?

*AC:*
*It is indeed possible that this decrease is an artefact from the spectral estimates, since log-smoothing uses less data points for lower frequencies and thus leads to higher spectral uncertainties there. We will add this remark to the manuscript here.*

P11, l1: 'In fact, the average correlation of 0.87 (0.94) over distances up to the maximal intercore distance...'
Please add here a reference to Appendix C, Figure C1. Otherwise the origin of these numbers (and their meaning) is unknown.
Based on the figure caption, this correlation is an average (for various intercore distances). Thus, maybe the formulation 'up to the maximum intercore distance' is misleading here. Why not say the correlation for average distance between cores instead?

*AC:*
*We apologize for the missing figure reference here. We will add the reference to Fig. C1, and include the average correlation values as symbols in this figure. In fact, the values are the average across all correlations for distances smaller than the maximum intercore distances, i.e. the average of all grey dots within the shaded regions in Fig. C1 (so not the correlation at the average core distance). We will rephrase the sentence to clarify this.*

P12, l. 11-13; 'The raw noise spectra derived from the two DML data sets exhibit a clear imprint from the diffusional smoothing in the firn, as suggested by the smaller PSD for periods <20 years of the raw DML2 spectrum, i.e. prior to correction, in comparison to DML1 (fig. 2b).'

This is too fast. Please split in two steps:

1. first, raw to corrected for both records (DML1 and DML2): the effect of diffusion is to decrease power at high frequencies;

2. second, comparison of the raw data for DML1 and DML2: DML2 is more affected by diffusion, since frequencies between 12 and 30 are affected (low power) only for DML2.

*AC:*
*We agree that the sentence condensed many pieces of information and might be hard to follow. We will thus split the sentence into several steps: "The raw noise spectra, i.e. prior to correction, derived from the two DML data sets exhibit a clear imprint from the diffusional smoothing in the firn. This is suggested by their common decrease in PSD towards shorter periods (Fig. 2b), since diffusion acts stronger on higher frequencies. It is corroborated by comparing the loss in PSD between the two data sets, which for DML2 is stronger towards the high-frequency end and also extends further towards longer periods. This is due to the stronger diffusional smoothing in the older sections of the cores that are only contained in the DML2 records, since the diffusion process had more time to act there since deposition of the snow."*

P12, l30: 'yields a slope of the DML2 signal spectrum of roughly beta=0.6'
Is it possible to add this fit to Figure 2 or 3?

*AC:*
*This would of course be possible, but in our oppinion it impairs the visual appearance of Fig. 2a which already contains many different line plots. Furthermore, sine the slope is here used only as a diagnostic means, we do not want to place special focus on its value. Therefore, we would rather refrain from adding the fit to Fig. 2a (neither to Fig. 3 since it shows a different quantity than discussed at this point of the manuscript).*

P13, l. 7: 'minimum averaging period constrained by the diffusion correction ( 2.5 years)'
Is it correct to compare r values at different averaging times (1 year and 2.5 years)?
Could this also contribute to the higher correlation compared to previous study of the same data?

*AC:*
*Of course you are right that we compare correlation values at slightly different averaging times, which also could contribute to the slight difference between the two values. We will add this as a remark here.*

P14, l. 1-3: 'For WAIS, the higher SNR at interannual timescales as compared to DML is consistent with the general notion that higher accumulation rates result in higher SNR.'

1. Please insert the SNR values for both.

*AC:*
*We will add the average SNR value for periods from 5-10 years for both DML and WAIS (as obtained from the data in Fig. 3) for better comparison here.*

2. For N=1 and deltat=5 years, r=0.45 at DML and r=0.51 at WAIS. The difference looks very small (especially considering the 3-times accumulation at WAIS). Why is the increase in SNR not scaled to the large increase in accumulation?

*AC:*
*Please note that we compare at this point the SNR values (Fig. 3) between the two regions and not the correlations in Fig. 4, since the latter cannot be compared one-to-one as the correlation values, which are currently shown in the figure, are based on different lower integration limits in Eq. (6) (DML lower limit of 500 yr period, WAIS lower limit of ~100 yr period), which is due to the different lengths of the data sets.*

*However, this review comment made us aware of the fact that it is more appropriate to use the same integration limits for both regions for Fig. 4. Thus, we will provide an updated version of the figure where we will use the same lower integration limits (i.e. 100 yr period), which we will also mention upon describing Eq. (6).*

P14, l26-28: 'Together with the observed spectral shapes on the longer timescales (30-100y) our results therefore might indicate a true increase in local variability at the WAIS sites towards longer time scales, and a close-to-constant or even decreasing coherent signal variability.'
This sentence is strange. It looks unbalanced with a first part dealing with long time scales and a second part dealing with... probably shorter time scales?
Is it possible to correct this sentence?
To facilitate reading, maybe this paragraph could be limited to the issue of high frequencies at WAIS, since the discussion for the long timescales already existed in the previous one.
Is it possible to separate the issue of high frequencies for noise and for signal?

*AC:*
*The sentence was intended to summarize both the findings on short and high frequencies: we argue before that deficiencies in the correction approach likely cannot explain the remaining decrease in noise level towards high frequencies. Taking into account additionally the results for the longer timescales, could then therefore suggest that the WAIS noise level tends to increase across the timescales studied and that the signal tends to stay constant or even decrease.*

*We suggest to clarify the sentence as follows: "By including the found spectral shapes of the signal and noise on the longer timescales (periods from 30-100 yr), together our results for the WAIS sites might therefore indicate that, across the timescales studied, there is a true increase in local variability and a close-to-constant or even decreasing coherent signal variability."*

What are the hypotheses explaining this decrease in noise at high frequencies? Could it be something like wind blowing that would homogenize only the surface over large areas? Or large-scale evaporation of surface snow homogenizing its composition?

*AC:*
*We see no obvious explanation for the decreasing noise at high frequencies and, given the uncertainty of our results, we would refrain here from suggesting such explanations which would be purely speculative.*

2.1 Appendix A

P16, l 23: 'a Gaussian convolution kernel whose standard deviation is the diffusion length $\sigma_i$ that is a function of depth (time) and depends on site i.'
This is very synthetic, but not very clear. Please add a reference to Appendix B where more details are provided. Please give some values of $\sigma_i$ (and possibly the associated frequencies) at various depths. Is there a formula that relates $\sigma_i$ to time (for each site)? If not, the use of the term 'function' might be misleading here. Indeed, $\sigma_i$ seems to answer on density, temperature and pressure, and therefore to be more similar to an adjustable parameter in the densification model.

*AC:*
*We will add the reference to Appendix B here and clarify the sentence as: "a Gaussian convolution kernel whose standard deviation is the diffusion length $\sigma_i$ that is a function of depth (or, equivalently: time) and depends on site i through depending on the local temperature, atmospheric pressure and accumulation rate (Appendix B)."*

*Please note that to our understanding it is correct to state that $\sigma_i$ is a function of depth: For constant temperature and pressure, the diffusion length is solely a function of firn density (Gkinis et al., 2014), and since density is a function of depth z, also $\sigma_i = \sigma_i(z)$. Assuming a Herron-Langway model for the firn density, there is an analytical solution which relates $\sigma_i$ to the firn density (van der Wel, 2012).*

*We will add some typical diffusion length values (both in cm and years) to the manuscript, which we think is most suitable in Appendix B (p.18 l.16).*

2.2 Appendix C

P19, l8: This sentence is complicated. It would be simpler to describe the procedure at EDML, and then say that the same method is applied to WAIS.

*AC:*
*We will rephrase the sentence as suggested.*

**Technical corrections:**

1. P5, l. 12: "inidividual" typo

*AC:*
*Thank you for spotting this typo which will be corrected.*

2. P6, l. 14: Please remove 'exemplarily' which is redundant with 'to illustrate our method'.

*AC:*
*We agree and will rephrase the sentence to: "In order to illustrate our method (Sect. 2.2), we first use the DML1 data set to demonstrate the individual steps involved in the analysis."*

3. P11, l. 8: 'estimated signal term' Please add a reference to the equation (5 or 6).

*AC:*
*We do not see a motivation for referencing Eq. 5 or 6 here, since these provide different quantities (i.e. the signal-to-noise ratio (SNR) and its integrated value). Instead we assume that the reviewer would welcome a reference to the relevant spectral quantities in the methods section 2.2 again for clarification. Since the term "estimated signal" has already been mentioned earlier in this paragraph, we suggest to provide this reference not here but directly in the first paragraph of the current section 4.1 (p.11, l.6-7): "We presented a method and the results of separating the variability recorded by Antarctic isotope records into two contributions: local variations ("noise", Eq. 4b) and spatially coherent variations ("signal", Eq. 4a)."*

4. P16, l. 12: 'core sites' typo

*AC:*
*Thank you for spotting this typo which will be corrected.*

5. P17, l.8: It is $F(\varepsilon_i)F(\varepsilon_j) \neq 0$ only for i=j, and hence... Maybe 'we have' instead of 'it is'?

*AC:*
*We will change the sentence as suggested.*

*Thomas Münch and Thomas Laepple*

References to the author comments:

Gkinis, V., et al., Earth Planet. Sci. Lett., 405, 132–141, https://doi.org/ 10.1016/j.epsl.2014.08.022, 2014.

van der Wel, L. G., Doctoral Thesis, University of Groningen, 146 pp., http://hdl.handle.net/11370/72cf3b0b-d258-44a1-8830-b0c355ddbd90, 2012.

---

## Author Comment (AC3) · 29 Oct 2018

Author Reply to the Editor Comment by Lukas Jonkers
on the manuscript

cp-2018-112: What climate signal is contained in decadal to centennial scale isotope variations from Antarctic ice cores?
by Thomas Münch and Thomas Laepple

Dear Lukas Jonkers,

below you find our answer to your comment on the above mentioned manuscript; your comment is set in normal font, our answers (author comment, AC) in *green italic font*.
* * *
Even though the manuscript is based on data that have already been published elsewhere, the following points will add to the reproducibility of your results:

- Please include a data availability statement, including data citations, for the data used (DML and WAIS ice cores and ERA Interim reanalysis data).

*AC:*
*We will include a data availability statement to the revised manuscript including the data citations for the used DML, WAIS and ERA-Interim data and add the respective references to the reference list.*

- Please consider to make code used in your study publicly available in online repository.

*AC:*
*We will add a code availability statement to the revised manuscript which will include a link to an online repository where the code used in our study will be made available.*

*Thomas Münch and Thomas Laepple*

---

## Author Response (ED1)

**Submission of revised version**

cp-2018-112: What climate signal is contained in decadal to centennial scale isotope variations from Antarctic ice cores? by Thomas Münch and Thomas Laepple

Dear Lukas Jonkers,

along with this document we hand in the revision of our manuscript entitled above.

In the revision, we have addressed all issues raised by both reviewers and the notes on the code and data availability mentioned by yourself. We have revised the respective manuscript parts as suggested in our reviewer responses; some changes slighlty differ from the ones suggested in the responses and for those we have added respective statements to the original response marked up in red. Emphasis in our revision was put on the Results section to add some more explanatory text in order to facilitate understanding by a broader audience of CP readers, as suggested by Reviewer 2. The final url and doi's for the code availability are in the process of being registered and will be added in the later process.

Please find attached the one-to-one response to the reviewer comments with all detailed changes we made, as well as a marked-up manuscript version created with latexdiff.

Thank you again for considering our manuscript.

On behalf of Thomas Lapple and with kind regards, Thomas Münch

**Author Reply to the Review Comments by Dmitry Divine (Referee #1)**

on the manuscript

cp-2018-112: What climate signal is contained in decadal to centennial scale isotope variations from Antarctic ice cores? by Thomas Münch and Thomas Laepple

Thank you very much, Dmitry Divine, for the time you spent on reading and reviewing our manuscript; we are happy about the positive outcome. Below we include a point-by point response to both the general and to all specific comments. The original referee comments are set in normal font, our answers (author comment, AC) are typeset in *green italic font*.

**Overall:**

In this manuscript the authors present a method for calculating a timescale dependent SNR for an array of climate proxy records with a common climatic signal and a physical mechanism(s) behind.

For a particular case presented, the study uses ice cores based isotopic records and accounts for the effects of stratigraphic noise, diffusion in firn and timescale uncertainties elaborating the respective power spectral densities for the background climate signal and the aforementioned contributing noise factors. The proposed technique is then applied to ice core arrays from DML and WAIS. Opposite timescale behavior of SNR for the two core networks is linked to the homogeneity/heterogeneity of distillation trajectories between the two regions associated, for example, with the effects of sea ice on isotopes in precipitation.

In general the paper is clearly written and results are well presented. Moreover, my general impression over this study, is that this was one of the rare cases I had so far as a reviewer that can be published almost "as is". When reading the manuscript, I left a number of remarks/suggestions/question marks that I planned to list later when writing this review, yet it turned out in the end that almost all of them the authors have already addressed in Discussion and Conclusions.

This study is certainly recommended for those who deals with multiproxy archives – this is an explicit demonstration of a value of a single proxy (ice core) record and a clear warning against overinterpreting single spikes/events on the shorter timescales. Therefore, I consider the manuscript deserves to be published after some very minor modifications to the content if the authors/editor finds them relevant.

**AC:**

Thank you very much for this positive evaluation of our work. We are happy that you find our study relevant, well presented and deserving publication.

**Minor Comments:**

Page 2, line 5: "...to a first approximation, changes in isotopic composition are only recorded in the ice if there is snowfall." Recent studies suggest the effects of air (and hence water vapor) exchange across the firn –air interface in between the precipitation events may have a larger impact on the final d18O in snow than previously thought, see for example Stenni et al., 2016, 10.5194/tc-10-2415-2016. It actually increases the role of SAT variability throughout the accumulation season even given the intermittency of precipitation itself.

**AC:**

We agree that there are indications that vapour exchange processes possibly further affect the isotopic composition of the surface snow and, if connected to SAT variations, could actually increase (again) the role of SAT variability for the isotope variability. Nevertheless, we could show for the Kohnen Station region in DML that, even after averaging away the local stratigraphic noise, the near-surface isotope variability is still very discrepant from the local interannual SAT varibility (Münch et al. 2017). This suggests either precipitation intermittency to be a main driver of the remaining (after reducing the

stratigraphic noise) isotope variability, or isotope modifications occurring directly at the surface which are not controlled by SAT variations.

In order to briefly reflect this discussion in the manuscript, we changed the respective sentence to: "To a first approximation, changes in isotopic composition are only recorded in the ice if there is snowfall, while the role of water vapour exchange processes in between precipitation events is still debated (Steen-Larsen et al., 2011; Stenni et al., 2016; Casado et al., 2018; Ritter et al, 2016; Münch et al, 2017)."

Page 6, line 4: Please provide a ref to eq. (7).

**AC:**

*The expression directly follows from the definition of the correlation coefficient as shown in Fisher et al. (1985). We added this reference to the paper here.*

Page 6, line 9: "... for display purposes... smoothed using a Gaussian kernel". Still the motivation is not clear, would it be possible to see an unsmoothed signal (in the letter of response for example). What is the kernel bandwidth used?

**AC:**

We agree that the formulation in the respective sentence was unclear. We do not apply the smoothing only for display purposes. In general, a strong smoothing is necessary in order to improve the quality of the spectral estimates. If one assumes that the spectrum of the climate signal (which one aims to reconstruct with proxy data) can be described by piecewise power laws, smoothing in logarithmic space is reasonable, which we have adopted here. This logarithmic smoothing is applied by taking weighted averages of spectral power over a Gaussian smoothing window with a scale factor in log units (Kirchner et al., 2005) which we choose to be 0.1 for the WAIS data and 0.15 for the DML data. The scaling in log units is proportional to the frequency at which the smoothing is applied, thus, at the higher frequencies more data points are averaged resulting in a stronger smoothing. We added a more thorough motivation for the logarithmic smoothing at this point of the paper.

Below you see an unsmoothed version of paper figure 1 where you can observe the strong spectral uncertainty when no smoothing is applied.

**Figure 1:** *As Fig. 1 from the paper but without logarithmic smoothing applied to the spectral estimates.*

Page 11, lines 10-20. Quality of ERA precipitation needs to be briefly discussed. How reliable are the estimates based on this variable?

**AC:**

We agree with the reviewer that the quality of the used ERA reanalysis data should be discussed but we think that the appropriate place for this would be the beginning of Appendix C. However, since our results do not critically depend on the accuracy of the estimated decorrelation scales, we included only a short and general discussion of the ERA-Interim quality of temperature and precipitation in Antarctica at the beginning of Appendix C.

Page 14, lines 30-33. The effects of sea ice on the modelled isotopic composition of precipitation in Antarctica can be found in the studies by Noone, e.g. Noone, D., and I. Simmonds (2004), Sea ice control of water isotope transport to Antarctica and implications for ice core interpretation, J. Geophys. Res., 109, D07105, doi:10.1029/2003JD004228. The authors are recommended to see if these results can be used to elaborate more on the potential controls of the different patterns in SNR found between the two study regions.

**AC:**

**Thank you very much for pointing us at this work. We included a reference to Noone and Simmonds and a short discussion of their findings at this point of the manuscript in order to improve our discussion of the role of sea ice changes for the isotopic composition of precipitation.**

Page 19, lines 3-5. The authors present the winter and summer precipitation results. It is highly recommended to do the same analysis for the fall and spring seasons. The semi-annual oscillation (SAO) tends to modulate the seasonal distribution of precipitation depending on the strength of the semiannual harmonic. In addition, for West Antarctica (though shown for Faraday only in Broeke et al., 2000, part 4) the sea ice extent in the Amundsen and Bellingshausen seas (also linked with SAO strength) was shown to modulate the seasonal precipitation too. One can speculate that a long term variability in the strength/position of the low in contraction phase of the SAO (March and September) can actually be one of the mechanisms responsible for disruption of the coherence between the isotopic records on the longer timescales.

See in the series of earlier publications by Van den Broeke.

**AC:**

Thank you very much for pointing us at these interesting results. We extended the decorrelation analysis of the ERA-Interim precipitation data for the fall and spring seasons. This, however, does not change the results: also for the distribution of spring or fall precipitation (e.g., spring minus fall or spring/total precipitation, etc.) the decorrelation scales lie roughly between 300 and 500 km for both regions. Thus, whatever season drives the precipitation intermittency, the typical spatial scale of the intermittency should be of the order of 300-500 km, and thus a factor of 3-4 below that of the temperature fields.

**Thomas Münch and Thomas Laepple**

**References to the author comments:**

Casado, M., et al., The Cryosphere, 12, 1745–1766, https://doi.org/10.5194/tc-12-1745-2018, 2018.

Fisher, D. A., et al., Ann. Glaciol., 7, 76-83, https://doi.org/10.1017/S0260305500005942, 1985.

Kirchner, J. W., Phys. Rev. E, 71, 066110, https://doi.org/10.1103/PhysRevE.71.066110, 2005.

Münch, T., et al., The Cryosphere, 11, 2175–2188, https://doi.org/10.5194/tc-11-2175-2017, 2017.

Ritter, F., et al., The Cryosphere, 10, 1647–1663, https://doi.org/10.5194/tc-10-1647-2016, 2016. Steen-Larsen, H. C., et al., J. Geophys. Res., 116, D06108, https://doi.org/10.1029/2010JD014311, 2011.

Stenni, B., et al., The Cryosphere, 10, 2415–2428, https://doi.org/10.5194/tc-10-2415-2016, 2016.

**Author Reply to the Review Comments by an anonymous reviewer (Referee #2)**

**on the manuscript**

cp-2018-112: What climate signal is contained in decadal to centennial scale isotope variations from Antarctic ice cores? by Thomas Münch and Thomas Laepple

We appreciate a lot the very thorough and detailed reading and reviewing of our mansucript by the anonymous reviewer. The many comments and suggestions will be of great help to improve our first version of the paper. Below we include a point-by point response to both the general and to all specific as well as technical comments raised by the referee. The original referee comments are set in normal font, our answers (author comment, AC) are typeset in *green italic font*.

**General comments:**

The article submitted deals with quantification of climate signal versus noise in ice cores from Antarctica. It is therefore well within the scope of Climate of the Past, and addresses an important issue for climatologists. Its aim is not to present new ice core data, but to present a methodology to evaluate (quantify) the climate signal contained in a series of records.

The methodology is based on a spectral analysis of the dataset, where the spectrum of the stacked record is compared to the mean spectrum and to white noise. The method also includes a correction for diffusion and for time uncertainty. The Methodology section is concise, because details are described in Appendixes. The paragraph 3.1 in Results is a useful complement to the methodology section, as it applies the method to an example, and provides a figure where the various steps are represented. It is well suited to an article that aims a large audience, not necessarily with statistical background, and who might overlook the equations in the methodology section and Appendixes.

**AC:**

We are happy that the structure of the manuscript and the steps at which we present the results are positively evaluated. Since not every reader of CP might be familiar with the details of the applied spectral analyses, it is indeed our intention to still present our results in a way that is understandable to a broader audience, as we envision our approach to be applicable in many fields of paleoclimatology/proxy research.

In the continuation of the Results section, the figures are described in less details. Some more precision is needed here, so that the main message is not obscured by unanswered questions on the parts of the figures that are not described. The results are different for the two studied regions. At EDML, the signal to noise ratio is found to increase for longer time scales (0.2 to 1), whereas at WAIS, it is relatively stable, and even seems to decrease at long (centennial) time scales. For the first region, the authors therefore recommend to use single cores only for multidecadal or longer timescales. For the second region, oppositely, they conclude that single cores yield good regional information at interannual and decadal scales, but give a more local information at longer time scales.

**AC:**

Thank you for these comments. Also in the light of what was said above, we edited the second part of the results section to ensure that the results are described in adequate precision in order to be comprehensible for a broad audience; see also our answers to the detailed comments.

In the Discussion, the authors consider the possible contribution of four processes to the climate signal, by looking at their spatial scale of coherence. They note that precipitation intermittency acts as noise or contributes to signal, depending on the scale considered.

They also discuss the unexpected decrease of signal power at WAIS for longer timescales. This coastal region is particularly sensitive to the variability in atmospheric circulation. They suggest that slow processes modifying the topography of the region may reduce the spatial coherence of the signal over long timescales.

The conclusions of the article are important regarding the confidence that we can attribute to one or several ice core records. The results for WAIS are unexpected, and therefore may trigger more research in the area, or allow to consider differently the results from previous drillings. Lastly, as noted by the authors, the new methodology can be applied to other records, and therefore be of interest to a large audience.

The manuscript is well written and well structured, with a good use of appendixes to procure relevant information. Given that the conclusions of the article are interesting, and well supported by the Method and Results sections, I recommend that this article should be accepted by Climate of the Past with minor corrections.

**AC: We appreciate this positive evaluation.**

**Specific comments:**

P2, 112: 'the diffusion of vapor through the firn column smoothes the isotope variations, reducing the power spectral density at high frequencies'

The first part of the sentence is very clear but the second part is less straightforward. Is it possible to add an intermediary step? '...smoothes the isotope variations, with stronger impact at short distances and therefore on the high frequencies.'

**AC:**

We will add the additional information, as suggested, and split the sentence into two parts for the sake of clarity: "Finally, once the snow is deposited, the diffusion of vapour through the firn column smoothes the isotope variations. This has a larger impact on short distances in the firn and therefore reduces the power spectral density of the variations strongest at the high frequencies (ref), which substantially shapes the isotope variability (ref)."

We changed the sentence to: "Finally, once the snow is deposited, the diffusion of vapour through the firn column smoothes the isotope variations (Johnsen, 1977; Whillans and Grootes, 1985). This has the strongest effect on short distances in the firn, significantly reduces the high-frequency power spectral density of the variations (Johnsen et al., 2000; van der Wel et al., 2015) and thereby substantially shapes the isotope variability (Laepple et al., 2018)."

P2, l. 24: 'Furthermore, knowing the timescale dependence, and thus the spectral shape of the noise, would allow the correction of the isotope-inferred variability estimates for the noise contribution.'

This sentence is complicated, it takes some time to find the subject. Is it possible to split it or reverse it?

'Furthermore, in order to correct the noise contribution to the isotope inferred variability estimates, it is necessary to know the noise distribution over various timescales, i.e. its spectral shape.'

**AC:**

We changed the respective sentence to: "Furthermore, in order to correct the isotope inferred variability estimates for the noise contribution, it is necessary to know the variance of the noise across timescales, i.e., its spectral shape."

P3, 114: 'The time uncertainty of the record chronologies, ..., has been reported with 2 % of the time interval to the nearest tie point.'

This sentence is unclear, is there something missing? Is it 'has been reported to be 2 % of the time interval'?

The sentence indeed was misleading and it should read "...reported to be 2% of the time interval". For the sake of clarity, we corrected this and also rearranged the sentence to: "The record chronologies were established from seasonal layer counting of chemical impurity records constrained by tie points from the dating of volcanic ash layers (Graf et al., 2002a). The resulting uncertainty of the chronologies has been reported to be 2 % of the time interval to the nearest tie point (Graf et al., 2002a), which ..."

**P5, 18: Equations 4**

Although it was only a few lines above, please repeat here what M and S are (respectively the mean of spectra, and the spectra of the stacked record along time). Also, is it possible to get another symbol for the noise (or at least a very specific font) so that it is easier to distinguish between N and N?

**AC:**

We repeated the definition of M and S here, as suggested, and changed the symbol for the number of records from upper case 'N' to lower case 'n' throughout the manuscript. We also note here that in Eq. (2), the lower limit of the sum was erroneously given by "i=0" instead of the correct "i=1". This was corrected.

P5, l. 19: 'we restrict our analyses to the frequency region where G<2. This avoids large uncertainties and yields cutoff frequencies of...'

Based on Figure B1 in annex, G is always below 2. Its maximum value is one. This seems logical since a filter will eliminate some frequencies, not add more frequencies to the signal. Should it be 'restrict to the region where G>0.5'? (so that the short frequencies will not triple in power after correction)?

Please clarify this point in the text or in the Appendix B.

Please define the term 'cut-off frequency' (and maybe draw the limits on Figure B1).

**AC:**

We apologize for the confusion. At this point of the manuscript we mixed up  $\overline{G}$  and  $\overline{G}^{-1}$  (the inverse of  $\overline{G}$ , which is actually applied as the correction function in Eqs. (4)). We will correct the sentence to "restrict to the region where  $\overline{G} \ge 0.5$ " and clarify in the appendix that this means a correction factor less than 2 in Eqs. (4). Additionally, we will define the phrase "cutoff frequencies" by paraphrasing the sentence as "This avoids large uncertainties and translates to a maximum frequency that is used for the analyses (hereafter: cutoff frequency) of... (Fig. B1)", and we will add the respective frequency limits as vertical lines in Fig. (B1a).

We revised the entire paragraph to clearly explain the approach.

**P6, l2: 'the value of F is related to the correlation between...'**

Is the correlation r providing more information than F (on the quality of the data) or the same information? Is the r value only for intercomparing with other studies?

**AC:**

The correlation is mathematically equivalent to the value of  $\overline{F}$ ; however, we think the correlation in addition provides a more direct and intuitive way of expressing the amount of variability contained in a stacked isotope record that is related to the common (climate) signal in relation to its total variability. In order to stress the equivalence of both quantities, we rephrased the sentence to "The value of  $\overline{F}$  can then be used to obtain the correlation between the time series of the common signal c and a "stacked" record  $\overline{x}$  built from the spatial average of n individual records:".

P6, 120-28:

1. Is it possible to insert here the symbols used previously in the methodology section (M, S) to facilitate the identification of the various terms?

2. The mean over all individual spectra, M, (figure1, black line)...

3. The mean spectrum divided by the number of records (M/15, figure 1, dashed line)...

4. ...averaging across record that contain noise and additionally... (S..., Figure 1, brown line)

**AC:**

We inserted the respective symbols here, as suggested.

Furthermore, please precise 'averaging across records along the time dimension' or 'in time space' to refer to the 'time stack' S, by opposition to the spectrum mean M.

**AC:**

We clarified the sentence as follows: "In comparison, averaging in the time domain across records that contain noise and additionally a common (i.e. spatially coherent) signal, ...".

P6, 129: 'For short timescales (2 to seven years), ..., is consistent with the null hypothesis of independent noise.'

The curves are closest between 3 and 5 years; they diverge again around 2. What causes this divergence from white noise? Is it an artefact?

**AC:**

This divergence from the white noise level close to the 2-yr Nyquist period is probably indeed an artefact caused by noise added in the measurement process, i.e. from the measurement uncertainty and more likely from the "jitter error", thus uncertainty in the definition of annual depth increments upon dating which translates into uncertainty of the annual averages. We added a respective remark to the manuscript here.

P7, 16-7: 'Unlike the average spectrum across all individual isotope variations (fig1), the corrected DML1 signal spectrum shows an increase of power spectral density with increasing time scale (fig. 2a)'.

1. Please precise which line is described here (color, symbol).

**AC: We will add the respective descriptions.**

2. Both the black and brown line show an increase of power spectral density towards higher timescales on Fig1 (strongest between 2 and 20, then flat area at longer timescales). This proposition has to be nuanced.

**AC:**

Please note that both these spectra in Fig1 have not been corrected for the loss of spectral power by diffusion, so most of the increase in spectral power between the 2 and 20 year period is attributable to the decreasing diffusional smoothing towards longer timescales. What we wanted to stress at this part of the manuscript is the difference in spectral shape between the mean spectrum (M, black line in Fig1) and the estimated signal (blue for DML1 in Fig2) for longer timescales, which stems from correcting the isotope variability for the noise contribution. We will rewrite the respective sentence to clarify this.

3. For the brown line, there is again an increase at very long timescales (>50) still on Figure 1.

**AC:**

This is indeed expected given that the estimated signal (blue line in Fig2), which is contained in the spectrum of the stack (see Eq. 3), increases in power towards the longer timescales.

4. On Figure 2, the 3 DML1 curves are very similar for t>20. So, the correction does not seem to have a huge effect. Moreover, the increase of power between 50 and 200 years exist with/ without correction...

**AC:**

It is indeed expected for the diffusion and time uncertainty corrections to have no effect for periods

>20 years, given their estimated transfer functions (Fig. B1). What is still relevant to correct on the longer timescales is the residual amount of noise still contained in the spectrum of the stack (since we only average a finite number of records). This residual noise is subtracted when calculating (solving for) the signal spectrum in Eq. (4a).

5. Lastly, regarding the increase between 3 and 20 years on Figure 2, the correction actually reduces the slope...

**AC:**

This is true and already noted in the manuscript (p.7 ll.11-13).

Possible correction to this sentence:

'Unlike the average spectrum across all individual variations (M, black line on Fig 1), the corrected DML1 signal spectrum (C, solid blue line on Figure 2a) shows an increase of power spectral densities with increasing time scales at time scales larger than 50 years.'

**AC:**

We added the respective descriptions, as suggested, and rewrote the paragraph to clarify that the corrections include both the correction for residual noise as well as diffusion+time uncertainty, and that they lead to a more steady increase in signal power as compared to the mean spectrum. For further clarification, we in addition changed the legend of the dashed line in Fig. 1a from "Raw" to "Uncorrected signal".

P7, l. 7-8: 'This is confirmed by the three 1000-yr long records from the DML2 data set whose signal exhibits a similar power spectrum in the range of timescales that overlap.' The two curves are not really a perfect match.

There is common behavior between 60 and 100 years (increasing) and between 5 and 15 years (increasing) but the middle part (15 to 60) is different between the two curves. DML1 is slowly decreasing (or flat), while DML2 is strongly decreasing until 30, then strongly increasing. Maybe add: '...for time scales longer than 50 years.' Or split the paragraph in two, one dealing with long time scales, and the other with short time scales.

**AC:**

We acknowledge that the two curves are indeed not a perfect match, but only show a similar slope. We weakened the statement accordingly.

P9, 1.6: '...decrease in signal power on centennial timescales'

Is it possible that this decrease is an artefact? How certain are we of the power of the 100-year frequency on a 200-year record?

**AC:**

It is indeed possible that this decrease is an artefact from the spectral estimates, since log-smoothing uses less data points for lower frequencies and thus leads to higher spectral uncertainties there. We added this remark to the manuscript here.

P11, 11: 'In fact, the average correlation of 0.87 (0.94) over distances up to the maximal intercore distance...'

Please add here a reference to Appendix C, Figure C1. Otherwise the origin of these numbers (and their meaning) is unknown.

Based on the figure caption, this correlation is an average (for various intercore distances). Thus, maybe the formulation 'up to the maximum intercore distance' is misleading here. Why not say the correlation for average distance between cores instead?

AC:

We apologize for the missing figure reference here. We added the reference to Fig. C1, and included the average correlation values as symbols in this figure. In fact, the values are the average across all correlations for distances smaller than the maximum intercore distances, i.e. the average of all grey

dots within the shaded regions in Fig. C1 (so not the correlation at the average core distance). We rephrased the sentence to clarify this.

P12, l. 11-13; 'The raw noise spectra derived from the two DML data sets exhibit a clear imprint from the diffusional smoothing in the firn, as suggested by the smaller PSD for periods <20 years of the raw DML2 spectrum, i.e. prior to correction, in comparison to DML1 (fig. 2b).'

This is too fast. Please split in two steps:

1. first, raw to corrected for both records (DML1 and DML2): the effect of diffusion is to decrease power at high frequencies;

2. second, comparison of the raw data for DML1 and DML2: DML2 is more affected by diffusion, since frequencies between 12 and 30 are affected (low power) only for DML2.

**AC:**

We agree that the sentence condensed many pieces of information and might be hard to follow. We thus split the sentence into several steps: "The raw noise spectra, i.e. prior to correction, derived from the two DML data sets exhibit a clear imprint from the diffusional smoothing in the firn. This is suggested by their common decrease in PSD towards shorter periods (Fig. 2b), since diffusion acts stronger on higher frequencies. It is corroborated by comparing the loss in PSD between the two data sets, which for DML2 is stronger towards the high-frequency end and also extends further towards longer periods. This is due to the stronger diffusional smoothing in the older sections of the cores that are only contained in the DML2 records, since the diffusion process had more time to act there since deposition of the snow."

P12, 130: 'yields a slope of the DML2 signal spectrum of roughly beta=0.6' Is it possible to add this fit to Figure 2 or 3?

**AC:**

This would of course be possible, but in our oppinion it impairs the visual appearance of Fig. 2a which already contains many different line plots. Furthermore, sine the slope is here used only as a diagnostic means, we do not want to place special focus on its value. Therefore, we would rather refrain from adding the fit to Fig. 2a (neither to Fig. 3 since it shows a different quantity than discussed at this point of the manuscript).

P13, l. 7: 'minimum averaging period constrained by the diffusion correction (2.5 years)'

Is it correct to compare r values at different averaging times (1 year and 2.5 years)? Could this also contribute to the higher correlation compared to previous study of the same data?

**AC:**

Of course you are right that we compare correlation values at slightly different averaging times, which also could contribute to the slight difference between the two values. We added this as a remark here.

P14, l. 1-3: 'For WAIS, the higher SNR at interannual timescales as compared to DML is consistent with the general notion that higher accumulation rates result in higher SNR.'

1. Please insert the SNR values for both.

**AC:**

We added the average SNR value for periods from 5-10 years for both DML and WAIS (as obtained from the data in Fig. 3) for better comparison here.

2. For N=1 and deltat=5 years, r=0.45 at DML and r=0.51 at WAIS. The difference looks very small (especially considering the 3-times accumulation at WAIS). Why is the increase in SNR not scaled to the large increase in accumulation?

AC:

Please note that we compare at this point the SNR values (Fig. 3) between the two regions and not the correlations in Fig. 4, since the latter cannot be compared one-to-one as the correlation values, which are currently shown in the figure, are based on different lower integration limits in Eq. (6) (DML lower limit of 500 yr period, WAIS lower limit of ~100 yr period), which is due to the different lengths of the data sets.

However, this review comment made us aware of the fact that it is more appropriate to use the same integration limits for both regions for Fig. 4. Thus, we provide an updated version of the figure where we will use the same lower integration limits (i.e. 100 yr period), which we also mention now upon describing Eq. (6).

Additionally, we added the statement that the interannual correlations on Fig.4 are only slightly different between the datasets since the correlation is based on the integrated SNR values.

P14, 126-28: 'Together with the observed spectral shapes on the longer timescales (30-100y) our results therefore might indicate a true increase in local variability at the WAIS sites towards longer time scales, and a close-to-constant or even decreasing coherent signal variability.'

This sentence is strange. It looks unbalanced with a first part dealing with long time scales and a second part dealing with... probably shorter time scales? Is it possible to correct this sentence?

To facilitate reading, maybe this paragraph could be limited to the issue of high frequencies at WAIS, since the discussion for the long timescales already existed in the previous one.

Is it possible to separate the issue of high frequencies for noise and for signal?

**AC:**

The sentence was intended to summarize both the findings on short and high frequencies: we argue before that deficiencies in the correction approach likely cannot explain the remaining decrease in noise level towards high frequencies. Taking into account additionally the results for the longer timescales, could then therefore suggest that the WAIS noise level tends to increase across the timescales studied and that the signal tends to stay constant or even decrease.

We suggest to clarify the sentence as follows: "By including the found spectral shapes of the signal and noise on the longer timescales (periods from 30-100 yr), together our results for the WAIS sites might therefore indicate that, across the timescales studied, there is a true increase in local variability and a close-to-constant or even decreasing coherent signal variability."

We changed the sentence to: "Taking additionally into account the spectral shapes we find for the signal and noise on the longer timescales (periods from  $\sim 30-100$  yr), our results might therefore overall indicate a true increase in local variability at theWAIS sites across the timescales studied and a close-to-constant or even decreasing coherent signal variability."

What are the hypotheses explaining this decrease in noise at high frequencies? Could it be something like wind blowing that would homogenize only the surface over large areas? Or large-scale evaporation of surface snow homogenizing its composition?

AC:

We see no obvious explanation for the decreasing noise at high frequencies and, given the uncertainty of our results, we would refrain here from suggesting such explanations which would be purely speculative.

**2.1 Appendix A**

P16, 123: 'a Gaussian convolution kernel whose standard deviation is the diffusion length  $\sigma_i$  that is a function of depth (time) and depends on site i.'

This is very synthetic, but not very clear. Please add a reference to Appendix B where more details are provided. Please give some values of  $\sigma_i$  (and possibly the associated frequencies) at various depths. Is there a formula that relates  $\sigma_i$  to time (for each site)? If not, the use of the term 'function' might be misleading here. Indeed,  $\sigma_i$  seems to answer on density, temperature and pressure, and therefore to be more similar to an

adjustable parameter in the densification model.

AC:

We added the reference to Appendix B here and clarified the sentence as: "a Gaussian convolution kernel whose standard deviation is the diffusion length  $\sigma_i$  that is a function of depth (or, equivalently, time since deposition) and depends on site i due to its dependence on local temperature, atmospheric pressure and accumulation rate (Appendix B)."

Please note that to our understanding it is correct to state that  $\sigma_i$  is a function of depth: For constant temperature and pressure, the diffusion length is solely a function of firn density (Gkinis et al., 2014), and since density is a function of depth z, also  $\sigma_i = \sigma_i(z)$ . Assuming a Herron-Langway model for the firn density, there is an analytical solution which relates  $\sigma_i$  to the firn density (van der Wel, 2012).

We added some typical diffusion length values (both in cm and years) to the manuscript, which we think is most suitable in Appendix B (p.18 l.16). In addition, we added some more details here on how we exactly calculate the diffusion lengths.

2.2 Appendix C

P19, 18: This sentence is complicated. It would be simpler to describe the procedure at EDML, and then say that the same method is applied to WAIS.

**AC:**

We rephrased the sentence as suggested.

**Technical corrections:**

1. P5, l. 12: "inidividual" typo

**AC:**

Thank you for spotting this typo which was corrected.

2. P6, l. 14: Please remove 'exemplarily' which is redundant with 'to illustrate our method'.

AC:

We agree and rephrased the sentence to: "In order to illustrate our method (Sect. 2.2), we first use the DML1 data set to demonstrate the individual steps involved in the analysis."

3. P11, l. 8: 'estimated signal term' Please add a reference to the equation (5 or 6).

**AC:**

We do not see a motivation for referencing Eq. 5 or 6 here, since these provide different quantities (i.e. the signal-to-noise ratio (SNR) and its integrated value). Instead we assume that the reviewer would welcome a reference to the relevant spectral quantities in the methods section 2.2 again for clarification. Since the term "estimated signal" has already been mentioned earlier in this paragraph, we provided this reference not here but directly in the first paragraph of the current section 4.1 (p.11, 1.6-7): "We presented a method and the results of separating the variability recorded by Antarctic isotope records into two contributions: local variations ("noise", Eq. 4b) and spatially coherent variations ("signal", Eq. 4a)."

4. P16, l. 12: 'core sites' typo

AC:

Thank you for spotting this typo was corrected.

5. P17, l.8: It is  $F(\varepsilon_i)F(\varepsilon_j) \neq 0$  only for i=j, and hence... Maybe 'we have' instead of 'it is'?

AC:

We changed the sentence as suggested.

Thomas Münch and Thomas Laepple

References to the author comments:

Gkinis, V., et al., Earth Planet. Sci. Lett., 405, 132-141, https://doi.org/ 10.1016/j.epsl.2014.08.022, 2014.

van der Wel, L. G., Doctoral Thesis, University of Groningen, 146 pp., http://hdl.handle.net/11370/72cf3b0b-d258-44a1-8830-b0c355ddbd90, 2012.

Author Reply to the Editor Comment by Lukas Jonkers on the manuscript

cp-2018-112: What climate signal is contained in decadal to centennial scale isotope variations from Antarctic ice cores?

by Thomas Münch and Thomas Laepple

Dear Lukas Jonkers,

below you find our answer to your comment on the above mentioned manuscript; your comment is set in normal font, our answers (author comment, AC) in *green italic font*.
* * *
Even though the manuscript is based on data that have already been published elsewhere, the following points will add to the reproducibility of your results:

- Please include a data availability statement, including data citations, for the data used (DML and WAIS ice cores and ERA Interim reanalysis data).

AC:

We included a data availability statement to the revised manuscript including the data citations for the used DML, WAIS and ERA-Interim data and add the respective references to the reference list.

- Please consider to make code used in your study publicly available in online repository.

AC:

We added code availability statements to the revised manuscript. Final url and doi's of the respective code repositories are in the process of being registered and will be added in the later process.

Thomas Münch and Thomas Laepple

[revised manuscript text omitted]

**3.2 Timescale dependence of DML and WAIS signal and noise variability**

After this detailed description for DML1, we now turn towards the results of the estimated signal and noise spectra for all three data sets. Unlike the average spectrum across all-

- 15 In general, the shape of the signal spectra is, as a result of the corrections, clearly distinct from the mean spectrum of the individual isotope variations(Fig. 1),. This is seen, for example, in the corrected DML1 signal spectrum shows an increase of power spectral density with increasing timescale which indicates a much more steady increase in PSD from short to long timescales (Fig. 2a)., solid blue line) as compared to the mean spectrum (Fig. 1, black line). This is confirmed by the three 1000 yr long records from the DML2 data set records whose signal exhibits a roughly similar power spectrum in the range of
- 20 timescales that overlap. This is partly expected We partly expect this since the longer cores are also part of the DML1 data set, but the increase in signal power in this frequency band seems to be a general feature over the entire last 1000 yr. In addition, the DML2 signal spectrum shows a further and similar increase for timescales beyond the 100 yr period. In the frequency band where the corrections for The change in spectral shape from the mean to the signal spectrum results from two contributions in the correction process: Firstly, the average isotope variability is corrected for the noise contribution ("uncorrected signal", the noise contribution of the noise contribution for the signal spectrum for the noise contribution for the none contribution for the
- 25 dotted lines in Fig. 2a). This correction is mostly apparent on the longer timescales leading to a higher slope in PSD of the signal spectrum compared to the mean spectrum (for DML1, the signal increases from the 10 to 100 yr period by a factor of  $\sim 2.6$ , the mean spectrum by a factor of  $\sim 1.4$ ). Secondly, the corrections for the loss in spectral power by the effects of diffusion and time uncertainty are important ((here important for time periods from  $\sim 20$  –to 5 yrperiods), the corrections cause.) lead to a smaller increase in PSD of the signal power spectra with increasing timescale as compared to the raw estimates (compare
- 30 dotted, dashed and solid lines in Fig. 2a). The corrected signal spectrum of the WAIS records In sharp contrast to DML, the corrections for the WAIS records yield a signal spectrum (Fig. 2c) is in stark contrast to the DML spectra, exhibiting, solid line) that exhibits a rather flat appearance throughout and indicating indicates decreasing PSD towards centennial timescales. Much of this the flat character is caused by the diffusion and time uncertainty correction which causes a strong amplification of corrections which strongly amplify the raw signal power for subdecadal timescales. on the subdecadal timescales; the decrease